# A One Health framework for exploring zoonotic interactions demonstrated through a case study

Amélie Desvars-Larrive [1,2] ✉, Anna Elisabeth Vogl [1], Gavrila Amadea Puspitarani [1,2], Liuhuaying Yang[2], Anja Joachim [3] & Annemarie Käsbohrer [1]

The eco-epidemiology of zoonoses is often oversimplified to host-pathogen interactions while findings derived from global datasets are rarely directly transferable to smaller-scale contexts. Through a systematic literature search, we compiled a dataset of naturally occurring zoonotic interactions in Austria, spanning 1975–2022. We introduce the concept of *zoonotic web* to describe the complex relationships between zoonotic agents, their hosts, vectors, food, and environmental sources. The zoonotic web was explored through network analysis. After controlling for research effort, we demonstrate that, within the projected unipartite source-source network of zoonotic agent sharing, the most influential zoonotic sources are human, cattle, chicken, and some meat products. Analysis of the One Health 3-cliques (triangular sets of nodes representing human, animal, and environment) confirms the increased probability of zoonotic spillover at human-cattle and human-food interfaces. We characterise six communities of zoonotic agent sharing, which assembly patterns are likely driven by highly connected infectious agents in the zoonotic web, proximity to human, and anthropogenic activities. Additionally, we report a frequency of emerging zoonotic diseases in Austria of one every six years. Here, we present a flexible network-based approach that offers insights into zoonotic transmission chains, facilitating the development of locally-relevant One Health strategies against zoonoses.

Zoonoses are caused by pathogens naturally transmissible between humans and wild or domestic animals. Places where humans and animals or animal products interact create interfaces that facilitate zoonotic agent transmission. Notably, approximately 99% of endemic zoonotic infections in humans originate from domesticated animals, within anthropogenic environments, either directly or indirectly through contaminated food or vectors[1]. Morand, et al.[2] provided statistical evidence supporting the positive relationship between the duration of domestication and the diversity of zoonotic agents that humans share with each domestic species, which was initially hypothesised by McNeill[3]. In addition, over 60% of human emerging infectious diseases (EIDs) are zoonotic[4]. Although direct zoonotic spillover from wildlife is rare and wildlife-to-human transmission typically occurs through indirect transmission[1], more than 70% of these zoonotic emergences are caused by pathogens with a wildlife origin[4]. However, the full host breadth of endemic and emerging zoonotic agents, as well as their animal and environmental reservoirs are rarely identified nor mapped.

[1]Centre for Food Science and Veterinary Public Health, Clinical Department for Farm Animals and Food System Science, University of Veterinary Medicine Vienna, Vienna, Austria. [2]Complexity Science Hub, Vienna, Austria. [3]Centre of Pathobiology, Department of Biological Sciences and Pathobiology, University of Veterinary Medicine Vienna, Vienna, Austria. ✉e-mail: amelie.desvars@vetmeduni.ac.at

In most zoonotic disease systems, interactions occur among multiple animal host species, environmental sources (including invertebrate vectors), and involve multiple infectious agents[5]. Therefore, exploring disease dynamics in these multi-source, multi-agent systems necessitates considering the complex ecology of the interactions, e.g., the host-pathogen community assemblages, the existence of environmental reservoirs, and the involvement of vectors[5-7]. Unfortunately, this complexity is often ignored due to the lack of comprehensive datasets, making it challenging to embrace a transdisciplinary perspective. Furthermore, network approaches to infectious diseases and spillover risk have largely focused on the analysis of the host-pathogen relationships[2,8-11], neglecting other sources of zoonotic infection, such as contaminated environment or food. A comprehensive understanding of circulating zoonotic agents, their hosts, vectors, food and environmental sources, and the key interfaces where spillover events may occur is essential for developing effective integrated One Health monitoring, prevention, and control of zoonoses[12].

Zoonotic and emerging diseases pose a significant threat to both human and animal health[13,14], they cause substantial economic losses[15], and may have far-reaching consequences on multiple aspects of the society[16]. The enhancement of monitoring efforts and data collection in both domestic and wildlife hosts is essential for effectively predicting the establishment of reservoirs, understanding the facilitators of zoonotic spillover, and preventing such spillover at source[17]. However, the ecology and diversity of circulating zoonotic agents are tied to multiple factors, including the local availability of potential animal hosts and vectors, their spatial distribution, density, population dynamics, and community composition[18,19]. In addition, the spillover force of infection depends on cultural and socio-economic determinants, including human agricultural practices, feeding and hunting habits, and behaviour[20,21]. This underscores the pressing need for the development of analytical tools to optimise surveillance strategies that are tailored to the regional or national context. While global datasets may be available[8,22-24], data granularity and completeness are generally suboptimal for smaller-scale investigations. Furthermore, there is a scarcity of national studies focusing on zoonotic interfaces that encompass animals, vectors, environmental, and food matrices. Bridging this gap is crucial for developing effective, locally relevant strategies[25] to monitor and mitigate potential changes in spillover risk that could impact human and animal health.

Austria has a growing population of nine million people. Its fauna encompasses approximately 45,870 species, of which 626 are vertebrates, including 110 mammalian and 418 avian species[26]. Moreover, of 3.9 million Austrian households, 35% own pets. The country also counts ~1.8 million cattle, 2.5 million pigs, and over 100 million poultry are slaughtered annually. Additionally, around 133,000 hunting permits are issued each year[27]. These numbers underline the importance of the human-animal interfaces at the national scale. Given the potential for zoonotic disease transmission at these interfaces and the ensuing risk to human health, Austria adheres to a combination of European and national regulations, guaranteeing a framework for coordinated epidemiological surveillance and responses. However, concentrating mostly on notifiable diseases, monitored and reported only for specific species, official figures tend to overlook non-regulated zoonotic agents circulating in the territory that could pose a risk to public health.

In this study, we extracted data from scientific papers and national laboratory reports spanning 47 years of publications, to generate a real-world network describing the web of zoonotic interactions in Austria and characterise the various interfaces through which zoonotic spillover may occur. We introduce the concept of "zoonotic web" (akin "food web")[28] as a network representation of zoonotic actors at human-animal-environment interfaces (i.e., [host-vector-environment-food]-zoonotic agent network), intended for use in One Health approaches. We treated it as a bipartite network and transformed it into a one-mode projection representing the network of zoonotic agent sharing among zoonotic sources, weighting relationships (edges) between zoonotic sources (nodes) by the number of zoonotic agents they shared. We explored this network using different network centrality metrics and a community-based approach. In addition, we examined zoonotic disease emergence patterns in Austria and pinpointed research trends and gaps on zoonotic agents at the national level.

## Results

### Dataset of zoonotic interactions in Austria, 1975–2022

The search identified 2186 publications. After 542 duplicates were removed, 1644 publications were screened with 1269 excluded at the title/abstract screening stage as they were not eligible (see Supplementary Fig. 1 for a breakdown based on exclusion criteria). This left 375 publications, of which 16 could not be retrieved, so 359 full-text articles were assessed for eligibility, with 229 meeting the criteria for final inclusion. In addition, 17 publications were found in excluded review articles, leading to a total of 246 publications that were ultimately included in this study (168 scientific articles, 13 reports, and 65 theses).

The final dataset is a *.csv. file with 2128 rows and 48 data fields. Each row represents one investigated zoonotic agent along with the results of the investigation in the animal host(s), vector(s), and environmental or food matrix(-ices). All included publications were published between January 1975 and August 2022. We evidenced a 17.8-fold increase in the number of publications on zoonoses in Austria between the first (1975–1997) and the second half (1998–2022) of the study period (Supplementary Fig. 2). In addition, there was variation in study distribution among federal states (Supplementary Fig. 3). To contextualise this result, it was compared with global data: a PubMed search using the terms (zoono* OR "zoono* disease*") from 1975 until 23 August 2022 (without restricting the search to Austria) generated a total of 64,282 results and revealed an increase of the same order (~ 18-fold). However, a PubMed search using the term "health" in the same period yielded 5,791,763 results, indicating a mere 6.8-fold increase in health publications globally. This result suggests a disproportionate rise in zoonotic disease research, both at the national and international levels, compared to general health studies.

### Research trends

Between 1975 and 2022, 227 unique zoonotic agents were investigated in Austria (not all of them could be resolved at species level). Ten genera collectively accounted for 41% of the selected literature: *Salmonella*, *Escherichia*, *Listeria*, *Echinococcus*, *Orthoflavivirus*, *Brucella*, *Toxoplasma*, *Campylobacter*, *Trichinella*, and *Leptospira* (Supplementary Table 1). Most zoonotic agents were studied in wildlife hosts, which accounted for 76.9% of the 221 animal species investigated. Furthermore, during the study period, the majority of investigations into food products concentrated on animal-origin products whereas plant-based foods (including fruits, vegetables, spices/herbs, and grains) accounted for 5.6% of the examined foodstuffs. Finally, across the selected publications, seven environmental matrices (including food and processing plants, public lavatory, sandbox, slaughter knife, soil, and water) and 21 invertebrate taxa (mosquitoes: 47.8%; ticks; 39.1%; sand flies, gastropods, and fleas: 4.3% each) were investigated.

In Austria, there has been a noticeable upward trend in scientific interest regarding zoonotic bacteria, viruses, and eukaryotes (Fig. 1a), with bacteria garnering the most attention. We observed an upward trend across all compartments, as recognised by the traditional One Health triad, i.e., animal, human, and environment, followed by a subsequent decrease in the number of studies investigating animals (from 2015) and humans (from 2010). The environmental aspect (including environmental media, plant-based food, and vectors) was not considered in studies on zoonotic diseases in Austria until 1997 but subsequently demonstrated the most gradual increase in scientific

interest (Fig. 1b), primarily driven by a rise in investigations on vectors (Supplementary Fig. 4).

## Zoonotic web actors and interfaces

Overall, between 1975 and 2022, the literature reported 197 zoonotic agents in Austria that were directly or indirectly evidenced in natural infections, including an unusual case of a dog hair described as a zoonosis (this "agent" was not considered in the network analysis) (Supplementary Fig. 5). Among them, 187 (94.9%) were directly or indirectly detected in 155 distinct vertebrate hosts, including human, 111 wildlife, eight livestock, and 36 companion animal (including exotic pets) species (Supplementary Table 2). The highest zoonotic agent richness was observed in Primates (88, with 87 zoonotic agents reported in humans), Carnivora (59 zoonotic agents), Artiodactyla (59), Galliformes (24), and Rodentia (23) (Fig. 2). In 78.6% (777 out of 989) of the positive results in hosts, direct detection was achieved and

represented the preferred method for bacteria and eukaryotes across all investigated host taxonomic classes. Conversely, viral circulation was primarily evidenced by indirect methods detecting antibodies (Table 1).

At the environment-zoonotic agent interface, 24 (12.2%) zoonotic agents were detected in 12 different invertebrate (vector) species. Surprisingly, despite the detection of the Usutu virus (USUV) in various bird species, horses, and humans across the reviewed studies, it was not reported in arthropod vectors, a necessary component of its biological cycle (Supplementary Table 3). In addition, 11 (5.6%) zoonotic agents, including bacteria (*Listeria monocytogenes*, *Salmonella* sp., *Escherichia coli*, and *Mycobacterium* sp.) and eukaryotes (*Cryptosporidium*, *Giardia*, and *Toxocara*) were reported in six types of environmental media, including surfaces and tools in food processing environments as well as "natural" matrices (e.g., water, sandboxes) (Supplementary Table 4). Finally, at the food system-zoonotic agent

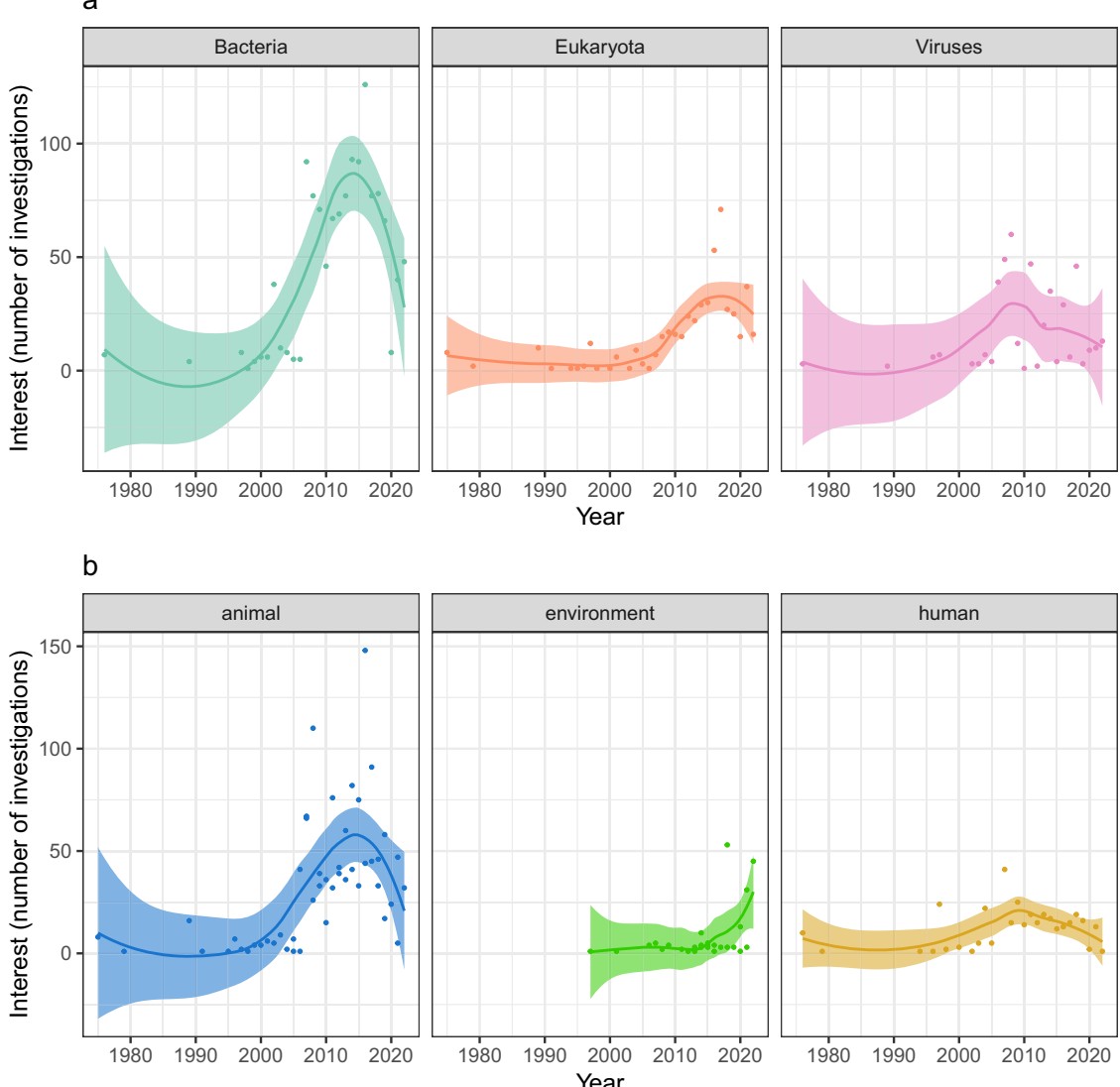

**Fig. 1 | Trends in research interest on zoonotic agents in Austria, 1975–2022. a** Trends in research interest measured by the number of investigations involving different superkingdoms of zoonotic agents. **b** Trends in research interest measured by the number of investigations involving each compartment, as recognised in the traditional One Health view. Dots represent the number of investigations per year; solid lines show a fitted trend (loess regression); shaded areas represent the corresponding 95% confidence interval. Only publications that investigated naturally occurring zoonotic infections were considered. Plant-based foodstuffs, invertebrate vectors, and any environmental matrices (including from food processing plants) were grouped under the compartment "environment" while food products of animal origin were considered within the "animal" compartment. Note that a single publication may present more than one investigation, i.e., investigating multiple zoonotic agents belonging to different superkingdoms and/or multiple compartments.

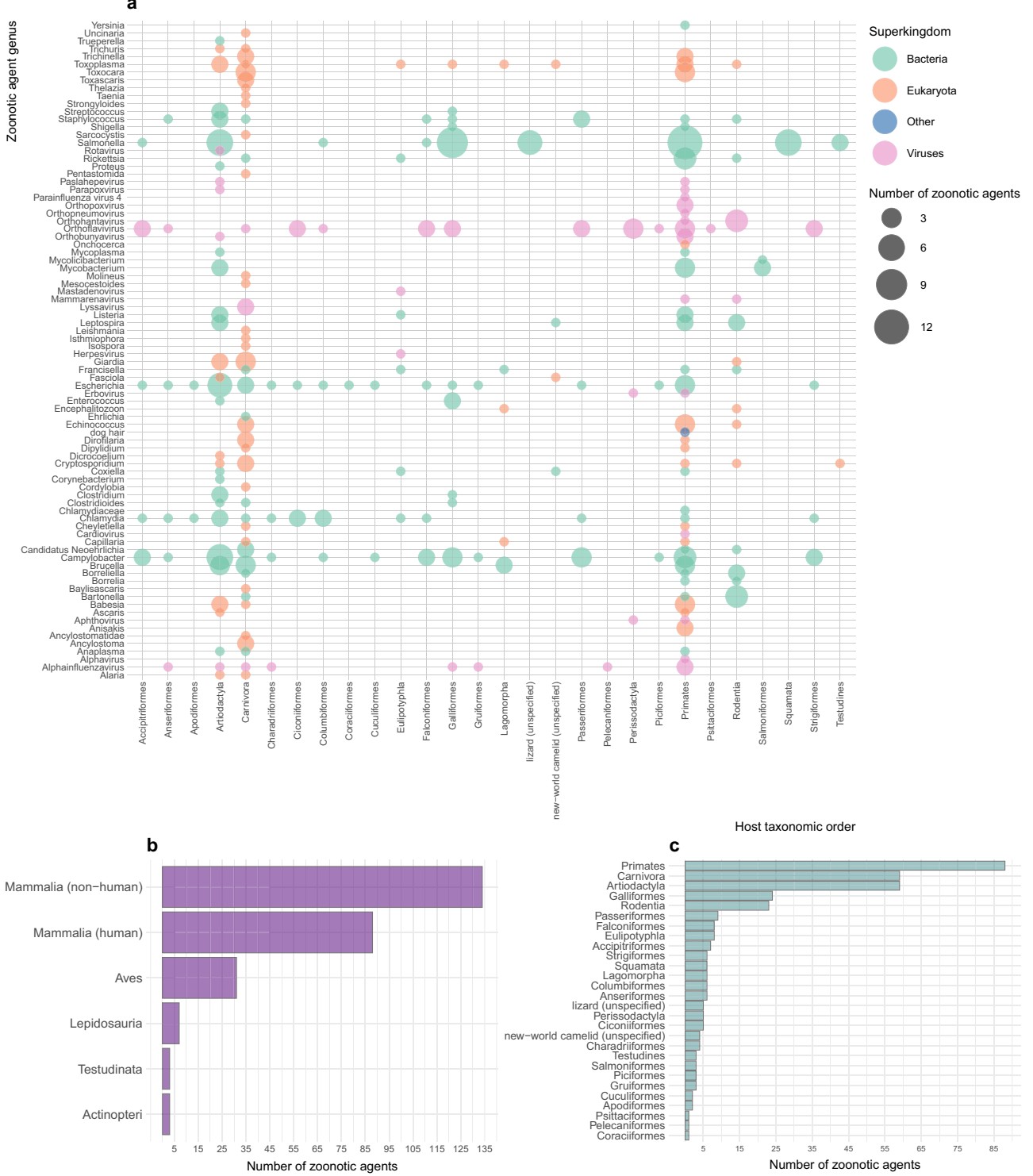

**Fig. 2 | Zoonotic agent distribution and richness across Austrian hosts, 1975–2022. a** Bubble plot illustrating the distribution of the zoonotic agent genera across Austrian hosts grouped by order, 1975–2022. Only publications that investigated naturally occurring zoonotic infections were considered. Bubble size corresponds to the number of zoonotic agents detected within a specific genus during this timeframe. Colours correspond to the zoonotic agent superkingdom. **b** Zoonotic agent richness per host taxonomic class, with data disaggregated for human and non-human mammals. **c** Zoonotic agent richness per host taxonomic order. When the host scientific or common name was not specific enough within the publication, the taxonomic order could not be retrieved, and the host name, as mentioned in the publication, was used (e.g., lizard, new-world camelid).

interface, 15 (7.6%) zoonotic agents were detected in 31 categories of food. Meat and meat products yielded the majority of positive results (55.8%) while plant-based foods comprised only 2.5% of zoonotic agent-positive food products. Zoonotic agents identified in food were mainly of the genera *Listeria* (36.6% of positive foodstuffs), *Escherichia* (22.8%), and *Salmonella* (22.5%). Out of the 21 identified zoonotic agents in foodstuffs, all were bacteria except for three parasites (*Anisakis, Echinococcus,* and *Trichinella spiralis*) (Supplementary Fig. 6).

**Table 1 | Breakdown of zoonotic agent detection methods showing the number of detections by host taxonomic class and zoonotic agent superkingdom, Austria, 1975–2022**

| Host taxonomic class | Zoonotic agent superkingdom | Detection method | Number of detections |
|---|---|---|---|
| Actinopteri | Bacteria | direct | 3 |
| Aves | Bacteria | direct | 104 |
| Aves | Bacteria | indirect | 1 |
| Aves | Bacteria | indirect and direct | 4 |
| Aves | Eukaryota | indirect | 1 |
| Aves | Viruses | direct | 40 |
| Aves | Viruses | indirect | 61 |
| Aves | Viruses | indirect and direct | 3 |
| Lepidosauria | Bacteria | direct | 48 |
| Mammalia (human) | Bacteria | direct | 154 |
| Mammalia (human) | Bacteria | indirect | 37 |
| Mammalia (human) | Bacteria | indirect and direct | 2 |
| Mammalia (human) | Eukaryota | direct | 46 |
| Mammalia (human) | Eukaryota | indirect | 18 |
| Mammalia (human) | Eukaryota | indirect and direct | 2 |
| Mammalia (human) | Viruses | direct | 9 |
| Mammalia (human) | Viruses | indirect | 21 |
| Mammalia (human) | Viruses | indirect and direct | 2 |
| Mammalia (human) | Other[1] | direct | 1 |
| Mammalia (non-human) | Bacteria | direct | 178 |
| Mammalia (non-human) | Bacteria | indirect | 30 |
| Mammalia (non-human) | Bacteria | indirect and direct | 13 |
| Mammalia (non-human) | Eukaryota | direct | 138 |
| Mammalia (non-human) | Eukaryota | indirect | 27 |
| Mammalia (non-human) | Eukaryota | indirect and direct | 11 |
| Mammalia (non-human) | Viruses | direct | 11 |
| Mammalia (non-human) | Viruses | indirect | 16 |
| Mammalia (non-human) | Viruses | indirect and direct | 3 |
| Testudinata | Bacteria | direct | 4 |
| Testudinata | Eukaryota | direct | 1 |

The class Mammalia is further disaggregated for humans and non-human mammals.
[1] This entry corresponds to the case when dog hair was evidenced under the skin of a human patient.

## Zoonotic web structure

Figure 3 depicts the zoonotic web and interfaces (see Supplementary Fig. 7 for conventional bipartite network visualisation). The network contained 396 nodes, i.e., actors (zoonotic sources and agents), with 658 edges (representing infections), and an average number of 1.66 interactions per actor. The giant connected component of the zoonotic web included 387 actors (97.7% of the nodes) with 652 edges (99.1% of the edges). In addition, the zoonotic web comprised three small components: the first illustrated relationships between *Encephalitozoon cuniculi* and its hosts *Arvicola amphibius* (Eurasian water vole) and *Oryctolagus cuniculus* (rabbit); the second showed *Mycobacterium chelonae*, *Mycobacterium marinum*, and *Mycolicibacterium fortuitum* with their common host *Salmo trutta fario* (river trout); finally, the third depicted the infection of mosquitoes of the genus *Uranotaenia* with Alphamesonivirus 1.

The analysis of the zoonotic web showed a right-skewed distribution of the node degree centrality (the number of links a node has), revealing few nodes with a high number of connections whereas most of the nodes had one. Among the hosts, the nodes *Homo sapiens* (human, degree centrality, $k = 87$), *Bos taurus* (cattle, $k = 38$), *Canis lupus familiaris* (dog, $k = 29$), *Felis catus* (domestic cat, $k = 21$), *Vulpes vulpes* (red fox, $k = 19$), *Sus scrofa* (pig, $k = 17$), *Gallus gallus* (chicken,

$k = 15$), *Ovies aries* (sheep, $k = 13$), *Sus scrofa (w)* (wild boar, $k = 11$), and *Nyctereutes procyonoides* (raccoon dog, $k = 10$) exhibited high zoonotic agent richness. Among the vectors, the node *Ixodes* exhibited the highest degree centrality ($k = 16$), with multiple connections to *Rickettsia*, *Borrelia*, and *Babesia* species. In contrast, the node *Culex* showed a low degree centrality ($k = 2$), with links to West Nile virus (WNV) and Orthobunyavirus Tahyna. Among nodes representing food sources, the highest degree centrality was observed for the nodes *cattle meat and meat product*, *animal (unspecified) meat and meat product* (each $k = 8$), and *animal (unspecified) dairy* ($k = 6$). The degree centrality of nodes representing environmental matrices showed relatively low values, ranging between 1 and 4. Among the zoonotic agents, the nodes USUV ($k = 38$), *Salmonella enterica* ($k = 33$), WNV ($k = 30$), *Salmonella* ($k = 24$), *Escherichia coli* ($k = 19$), *Listeria* ($k = 17$), *Listeria monocytogenes* ($k = 17$), verotoxigenic *Escherichia coli* (VTEC) ($k = 16$), *Campylobacter jejuni* ($k = 15$), *Toxoplasma gondii* ($k = 15$), Influenza A virus ($k = 12$), *Campylobacter coli* ($k = 11$), enterohaemorrhagic *E. coli* (EHEC) ($k = 11$), *Leptospira*, *Staphylococcus aureus*, and *Campylobacter* (each $k = 10$) revealed a greater zoonotic source plasticity (range) in Austria. Furthermore, with an average degree of 3.90, viruses had a greater zoonotic source plasticity than bacterial (3.77) or eukaryotic (2.28) zoonotic agents.

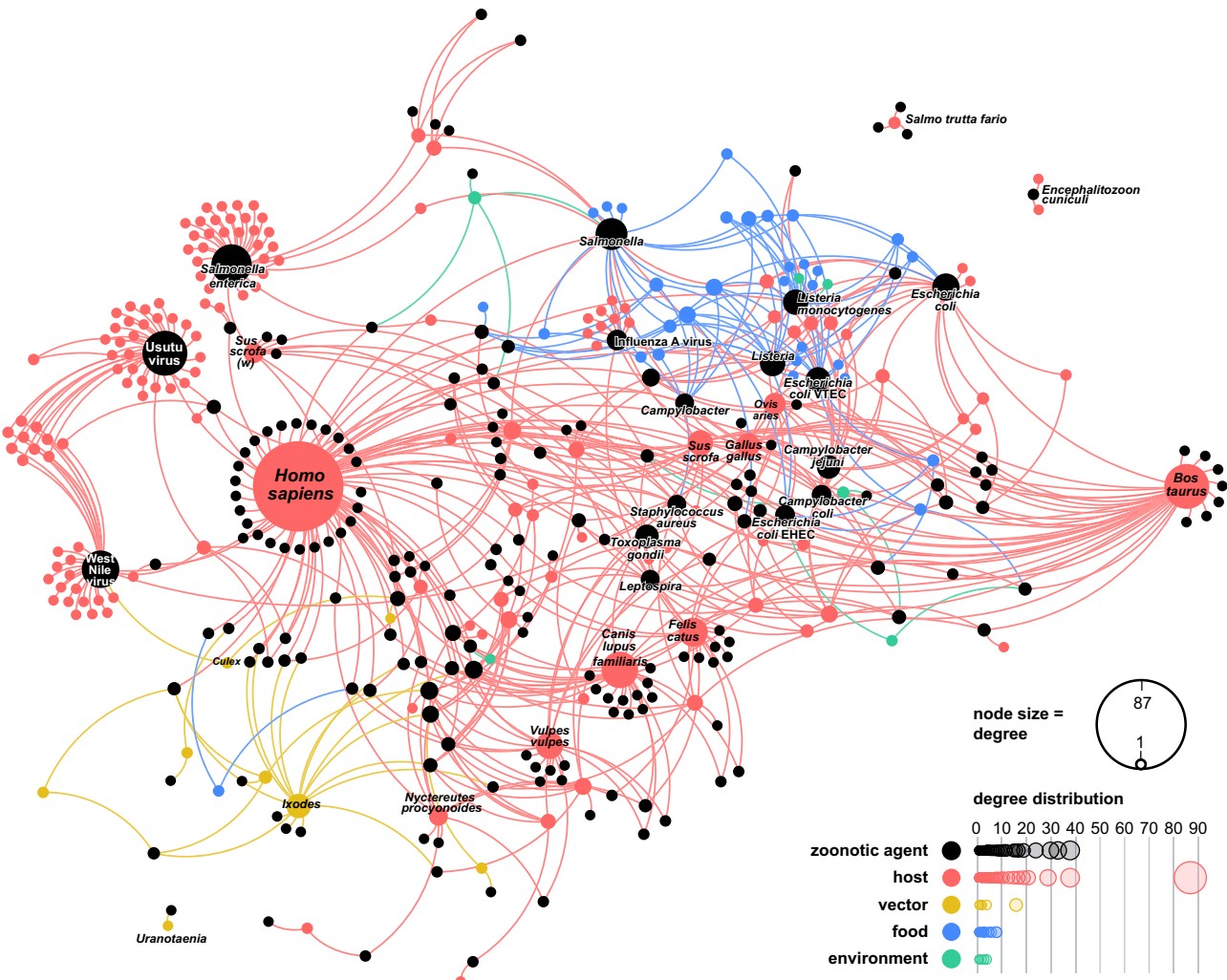

**Fig. 3 | Network representation of the zoonotic web in Austria, 1975–2022.** This representation uses the D3 forceLink layout, providing a detailed visualisation and offering epidemiological insights into naturally occurring zoonotic interactions in Austria. The zoonotic web is a bipartite network, where each node (circle) represents an actor in the zoonotic web, with one set of nodes representing zoonotic agents (black nodes) and the second set representing zoonotic sources that belong to different categories: vertebrate hosts (red nodes), vectors (yellow nodes), foodstuffs (blue nodes), and environmental matrices (green nodes). A link between a zoonotic agent *i* and a vertebrate host *j* indicates that agent *i* was directly or indirectly detected in host *j*; a link between a zoonotic agent *i* and a vector *j* signifies that agent *i* was identified in vector *j*, implying that vector *j* may transmit agent *i* to a vertebrate host through a bite or mechanically; a link between a zoonotic agent *i* and an environmental matrix *j* indicates the presence of agent *i* in environment *j*, potentially leading to infection of a vertebrate host upon contact; and a link between a zoonotic agent *i* and a food matrix *j* indicates that agent *i* was detected in food *j*, which may result in the infection of a vertebrate host through ingestion. Node size represents the actor's degree centrality. The node degree centrality for each zoonotic source corresponds to the zoonotic agent richness, i.e., the number of taxa directly or indirectly evidenced from the zoonotic source. The node degree centrality for each zoonotic agent corresponds to the zoonotic source range, i.e., the number of sources from which the agent has been directly or indirectly evidenced, reflecting its "host" or "zoonotic source" plasticity. The bottom-right graph illustrates the degree distribution for the "zoonotic agents" and "zoonotic sources" partitions, the latter being disaggregated based on source categories. Interactive version at: https://vis.csh.ac.at/zoonotic-web/dashboard.html.

## Network of zoonotic agent sharing

We generated a unipartite scientific research effort-adjusted network of zoonotic sources (i.e., accounting for research biases), based on zoonotic agent sharing. This network depicts patterns of zoonotic transmission potential between sources, with edges representing the likelihood that a given zoonotic source will transmit one or more zoonotic agents to another source relative to other sources in the network[29]. Thus, for one zoonotic agent, connected sources belong to the same potential transmission chain[29,30] (Fig. 4a). In this network, node rankings using the four centrality metrics (degree; strength, i.e., the sum of the weights of edges to/from a node; betweenness, i.e., the number of shortest paths that go through a node; and closeness, i.e., the average distance to all other nodes)[31] showed positive correlation (0.26 < Kendall's Tau < 0.77, *p* < 0.001 in all cases, Supplementary

Table 5). Degree and strength centrality reflect co-occurrence patterns of zoonotic agents among sources[9]. In contrast, betweenness and closeness centrality provide insights into indirect interactions through other sources[32]. The nodes *Homo sapiens* (human), *Gallus gallus* (chicken), *Bos taurus* (cattle), and *animal (unspecified) meat and meat product* were the most influential nodes in the network, appearing in the top 10 actors by the four centrality metrics. In addition, the nodes *Ovies aries* (sheep) and *cattle meat and meat product* could also be considered influential, ranking in the top 10 actors by three (out of four) centrality metrics (Table 2). Notably, the nodes *Equus caballus* (horse) and various nodes representing bird species exhibited high degree and strength centrality, attributable to their shared interactions with the two Orthoflaviviruses, WNV and USUV. Interestingly, the nodes *Sus scrofa* (wild boar), *Testudines* (turtles), *Canis lupus familiaris*

(dog), *Felis catus* (domestic cat), *Apodemus flavicollis* (yellow-necked field mouse), *Nyctereutes procyonoides* (raccoon dog), and the tick *Ixodes* ranked high by betweenness centrality, suggesting that they may act as bridges between host communities[33,34]. Besides the most

influential nodes, two hosts, *Sus scrofa* (pig) and *Canis lupus familiaris* (dog), as well as two food matrices, *animal (unspecified) ready to eat product* and *pig meat and meat product,* ranked in the top 10 actors by closeness centrality. Closeness centrality identifies nodes that are

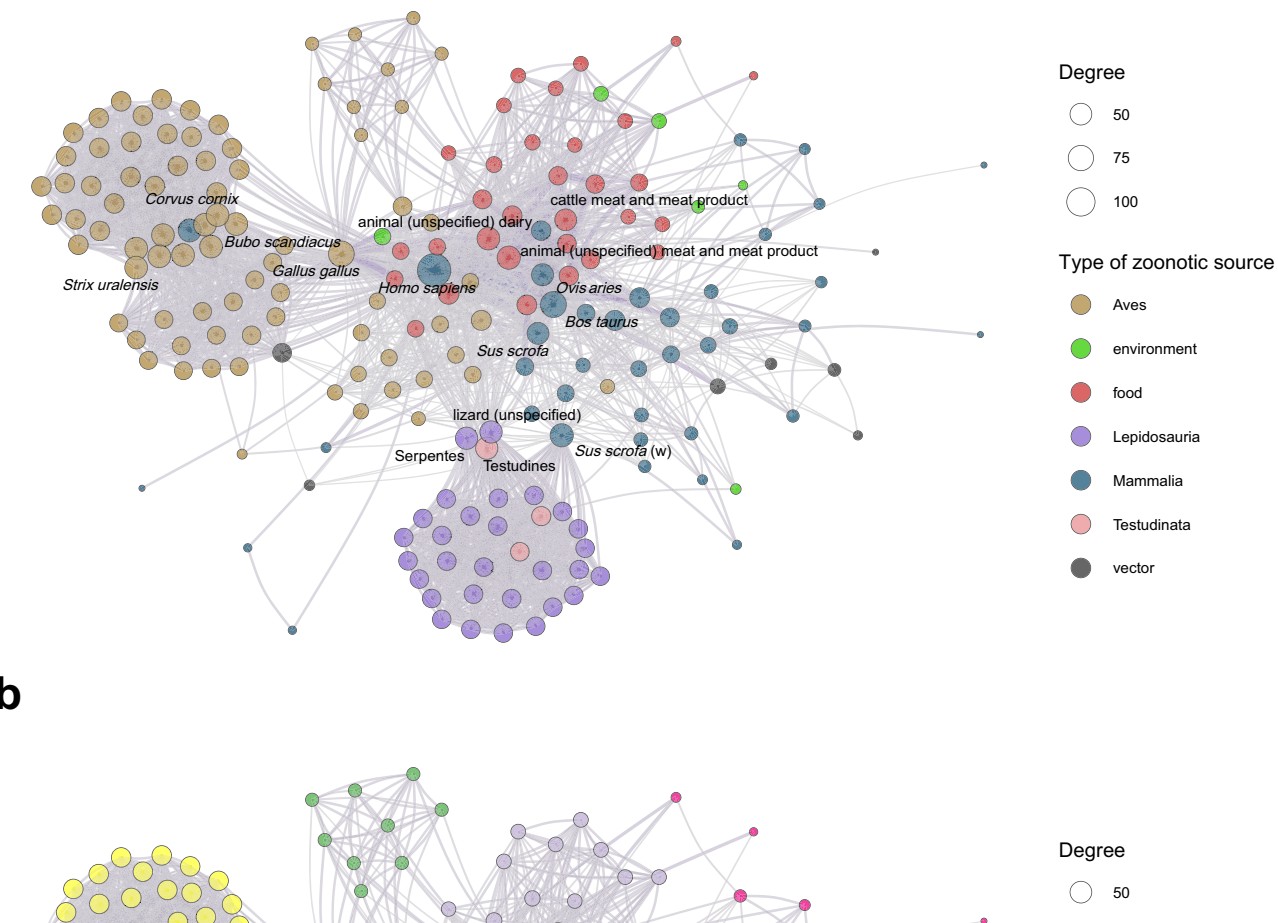

**Fig. 4 | Network of zoonotic agent sharing, created by connecting two zoonotic sources when they share at least one zoonotic agent.** The network is unipartite, and each node (circle) represents a zoonotic source that belongs to a source category: vertebrate host, vector, food, or environmental matrix. Node size represents the zoonotic source's degree. The weight (width) of an edge between two zoonotic sources represents the number of zoonotic agents shared, adjusted for the scientific research effort. **a** Transmission-potential network among zoonotic sources. Node colours depict zoonotic source categories. Zoonotic hosts are additionally colour-coded based on taxonomic classes to offer further biological insights. **b** Communities of zoonotic sources based on zoonotic agent sharing as determined using the Leiden algorithm. Node colours represent the communities.

**Table 2 | Top 10 most influential actors in the network of zoonotic agent sharing (i.e., considering hosts, vectors, food, and environmental matrices as zoonotic sources) ranked by node centrality metrics**

| Node / Degree centrality | Node / Strength centrality | Node / Betweenness centrality | Node / Closeness centrality |
|---|---|---|---|
| 1. *Homo sapiens* (human)[1] / 149 | 1. *Homo sapiens* (human)[1] / 993.3 | 1. *Homo sapiens* (human)[1] / 0.732 | 1. *Homo sapiens* (human)[1] / 3.92 |
| 2. *Bos taurus* (cattle)[1] / 79 | 2. *Bos taurus* (cattle)[1] / 457.3 | 2. *Sus scrofa* (wild boar)[1] / 0.212 | 2. *Bos taurus* (cattle)[1] / 3.70 |
| 3. *Gallus gallus* (chicken)[1] / 78 | 3. *Gallus gallus* (chicken)[1] / 387.2 | 3. *Bos taurus* (cattle)[1] / 0.122 | 3. *Gallus gallus* (chicken)[1] / 3.61 |
| 4. *Sus scrofa* (wild boar)[1], Animal (unspecified) meat and meat product[3] / 59 | 4. Animal (unspecified) meat and meat product[2] / 383.7 | 4. *Gallus gallus* (chicken)[1] / 0.100 | 4. Animal (unspecified) meat and meat product[2] / 3.54 |
| 5. *Equus caballus* (horse)[1] / 57 | 5. Cattle meat and meat product[2] / 281.0 | 5. Testudines (turtles)[1] / 0.025 | 5. *Sus scrofa* (pig)[1] / 3.51 |
| 6. *Asio otus* (long-eared owl), *Bubo bubo* (Eurasian eagle-owl), *Bubo scandiacus* (snowy owl), *Ciconia Ciconia* (white stork), *Circus aeruginosus* (western marsh harrier), *Coloeus monedula* (jackdaw), *Corvus cornix* (hooded crow), *Gypaetus barbatus* (lammergeier), *Strix uralensis* (Ural owl)[1], animal (unspecified) dairy[2] / 56 | 6. *Ovis aries* (sheep)[1] / 270.4 | 6. *Canis lupus familiaris* (dog)[1] / 0.018 | 6. *Ovis aries* (sheep)[1] / 3.46 |
| 7. Serpentes, Testudines, lizard (unspecified)[1] / 53 | 7. Animal (unspecified) dairy[2] / 265.1 | 7. *Felis catus* (domestic cat)[1] / 0.016 | 7. Cattle meat and meat product[2] / 3.42 |
| 8. *Ovis aries* (sheep)[1] / 51 | 8. *Gypaetus barbatus* (lammergeier)[1] / 259.7 | 8. Animal (unspecified) meat and meat product[2] / 0.013 | 8. Animal (unspecified) ready to eat product[2] / 3.39 |
| 9. *Sus scrofa* (pig)[1], cattle meat and meat product[2] / 50 | 9. *Bubo scandiacus* (snowy owl), *Circus aeruginosus* (western marsh harrier), *Coloeus monedula* (jackdaw), [1] / 256.6 | 9. *Apodemus flavicollis* (yellow-necked field mouse), *Nyctereutes procyonoides* (raccoon dog)[1] / 0.0103 | 9. Pig meat and meat product[2] / 3.38 |
| 10. Game meat and meat product[2] / 44 | 10. *Strix uralensis* (Ural owl)[1] / 256.3 | 10. *Ixodes*[3] / 0.101 | 10. *Canis lupus familiaris* (dog)[1] / 3.32 |

The NCBI-resolved scientific and common names of the hosts are specified. Edge weights were adjusted to take into account the scientific research effort. The normalised values of the weighted betweenness and closeness are presented.
[1] Zoonotic source category: host; [2] Zoonotic source category: food; [3] Zoonotic source category: invertebrate vector.

**Table 3 | Summary statistics of the node centrality metrics in the research effort-adjusted network of zoonotic agent sharing, per category of zoonotic sources**

| | Min. | 1st Qu. | Median | Mean | 3rd Qu. | Max. |
|---|---|---|---|---|---|---|
| **Hosts (n = 152)** | | | | | | |
| Degree centrality | 1 | 19 | 32 | 30.05 | 37 | 149 |
| Strength centrality | 1.58 | 58.86 | 155.18 | 139.66 | 171.23 | 993.30 |
| Betweenness centrality | 0 | 0 | 0 | 0.0087 | 0.0007 | 0.731 |
| Closeness centrality | 0.877 | 1.852 | 2.158 | 2.196 | 2.466 | 3.919 |
| **Vectors (n = 7)** | | | | | | |
| Degree centrality | 1 | 4 | 7 | 10.86 | 13.5 | 33 |
| Strength centrality | 3.88 | 7.235 | 19.85 | 35.53 | 40.04 | 130.41 |
| Betweenness centrality | 0 | 0 | 0 | 0.0015 | 0.0030 | 0.0101 |
| Closeness centrality | 1.131 | 1.508 | 1.674 | 1.853 | 2.076 | 2.995 |
| **Food (n = 31)** | | | | | | |
| Degree centrality | 2 | 16 | 23 | 25.68 | 33 | 59 |
| Strength centrality | 10.24 | 73.58 | 89.19 | 128.76 | 165.90 | 383.69 |
| Betweenness centrality | 0 | 0 | 0 | 0.0005 | 0.0002 | 0.0134 |
| Closeness centrality | 1.676 | 2.255 | 2.583 | 2.654 | 3.159 | 3.542 |
| **Environment (n = 6)** | | | | | | |
| Degree centrality | 3 | 6.25 | 13 | 12.17 | 16 | 23 |
| Strength centrality | 11.63 | 22.90 | 52.78 | 48.75 | 70.75 | 85.62 |
| Betweenness centrality | 0 | 0 | 0 | 0 | 0 | 0 |
| Closeness centrality | 1.928 | 1.973 | 2.067 | 2.113 | 2.278 | 2.324 |

"close" to many other nodes[31]; therefore, zoonotic sources which share numerous zoonotic agents with numerous sources would have high closeness centrality[33]. Summary statistics for the four-node centrality metrics per category of zoonotic sources are shown in Table 3. Many nodes in the network showed a betweenness equal to zero. Except for betweenness centrality, there were significant differences in the average values of the centrality metrics between the four zoonotic source categories (Supplementary Table 6).

**Zoonotic agent sharing communities**

We identified six communities (clusters of zoonotic sources sharing similar agents) in the zoonotic agent sharing network (Fig. 4b). Community 1: primarily comprised of central hosts having higher values of centrality in the unipartite zoonotic agent sharing network and generally living in proximity to humans or having frequent interactions with humans, including livestock, companion animals (dog, cat), synanthropic species (Norway rat, domestic mouse), game species (red fox, cervids), but also captive primates. Notably, *Aedes* mosquitoes and ticks (*Hyalomma*, *Ixodes*) were part of this community. Community 1 was characterised by a high diversity of zoonotic agents, with 175 taxa shared among 51 zoonotic sources that composed the community. Community 2: encompassed diverse reptiles (snakes, lizards, and turtles) and amphibians, including non-traditional pet (NTP) species, along with the wild boar; the main zoonotic agent shared within this community was *S. enterica*. Community 3: consisted of various avian taxa, including birds of prey, ducks, waterfowl, gamebirds, chickens, and pigeons. Note that hosts in this community were broadly designated, lacking specific scientific nomenclature. The primary shared zoonotic agents in community 3 were *E. coli* and Influenza A virus. Community 4: included various food products and environmental matrices related to food production, but also public lavatory and *Meleagris gallopavo* (turkey). The main zoonotic agents shared within community 4 were foodborne, principally *Salmonella*, *Listeria monocytogenes*, and VTEC. Community 5: mostly clustered

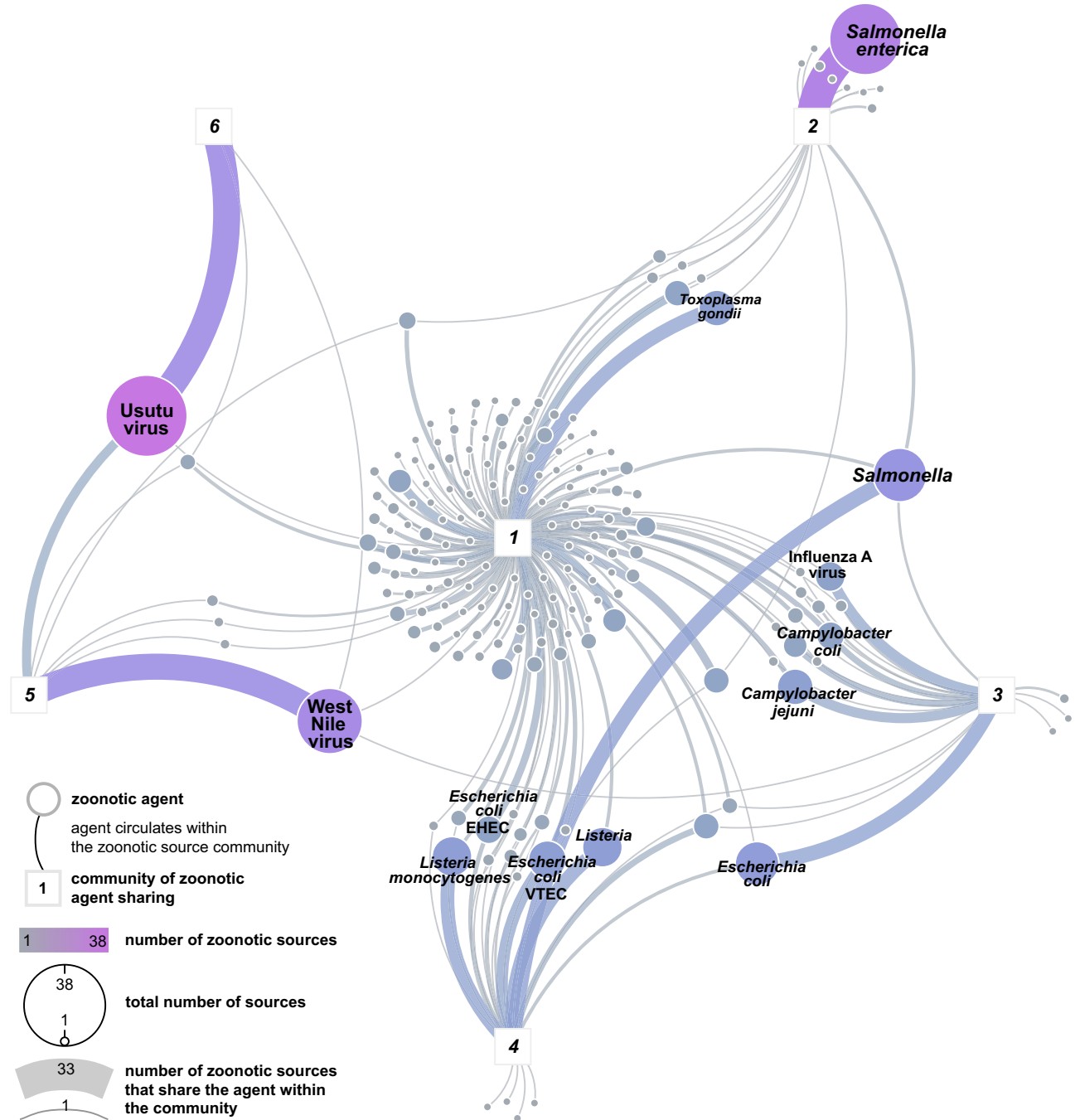

**Fig. 5 | Shared zoonotic agents within each zoonotic source community.**
Communities (represented by squares) were determined using the Leiden algorithm run on the research effort-adjusted zoonotic agent sharing network. Circles represent zoonotic agents. The size of each circle represents the degree centrality of the node in the bipartite zoonotic web (i.e., the total number of sources from which it has been directly or indirectly evidenced, reflecting its "host" or "zoonotic source" plasticity). The circulation of a zoonotic agent within a community is represented by a link between the community (square) and the zoonotic agent (circle). Link width represents the number of zoonotic sources that share the zoonotic agent within the community it is linked to. The colour scale shows the number of zoonotic sources (the colour scale is correlated to both node size and link width). Interactive version at: https://vis.csh.ac.at/zoonotic-web/dashboard.html.

WNV hosts and, to a lesser extent, USUV hosts, including various bird species, the vector *Culex*, and horses. Community 6: represented USUV hosts and exclusively included bird species (Fig. 5, Supplementary Table 7 and Supplementary Fig. 8).

### Zoonotic agent sharing at human-animal-environment interfaces

A total of 24,475 3-cliques were identified, of which 153 were One Health 3-cliques. The distribution of the research effort-adjusted

number of zoonotic agents shared at human-animal-environment interfaces (represented by the sum of the edge weights within One Health cliques) displayed a right-skewed pattern (Fig. 6a), with a median of 20.8. This suggested that, at most human-animal-environment interfaces, the likelihood of a specific zoonotic source transmitting one or more zoonotic agents to another source is relatively low. We identified six One Health cliques that ranked the highest based on the number of zoonotic agents shared (Fig. 6b). In five of them, cattle (*B. taurus*) was involved, while in two of them, foodstuffs

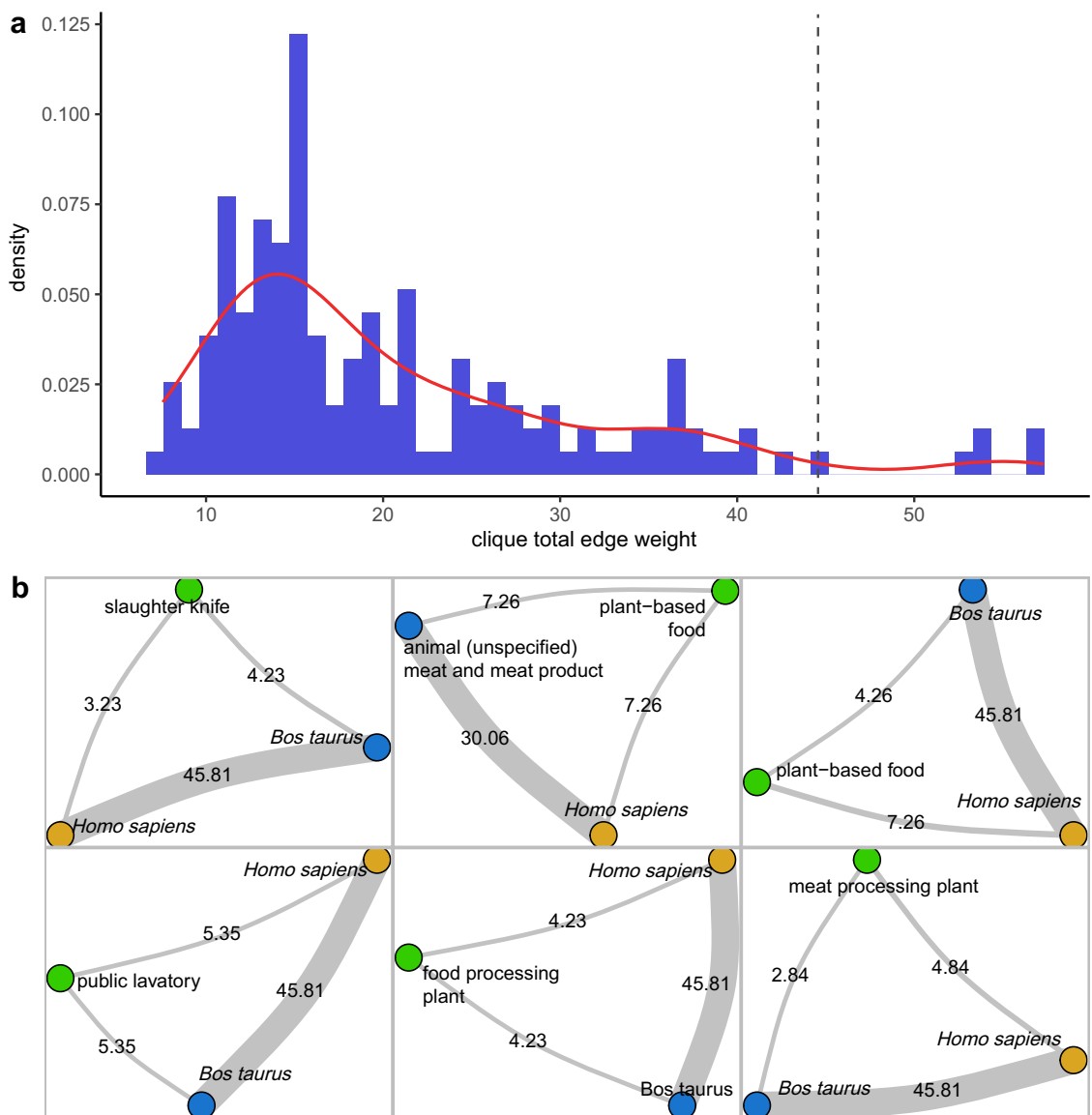

**Fig. 6 | Analysis of the One Health 3-cliques in the zoonotic agent sharing network, Austria, 1975–2022. a** Histogram and density plot of the total edge weight in One Health 3-cliques within the network of zoonotic agent sharing. The edge weight between two nodes (representing zoonotic sources belonging to different One Health compartments) is the number of zoonotic agents shared between these two nodes, adjusted for the scientific research effort. The total edge weight of a clique corresponds to the sum of the edge weights between all pairs of nodes within the clique. Plant-based foodstuffs, invertebrate vectors, and any environmental matrices (including from food processing plants) were included into the compartment "environment" while food products of animal origin were considered within the "animal" compartment. The dashed line separates the top six cliques (right side) from the others. **b** Top six One Health 3-cliques based on the total edge weight. Nodes are colour-coded based on the "traditional" One Health compartment, with yellow representing the human compartment, blue representing the animal compartment, and green representing the environment. Edge weight is visualised as the edge width and corresponding edge label.

from animal or plant origin were implicated. Environmental samples from food processing facilities were present in three cliques.

### Imported and emerging zoonotic agents

Between 1975 and 2022, Austria reported 48 importation events of zoonotic agents, of which, 11 were bacteria, seven were helminths, one was an arthropod, and two were viruses. These imported zoonotic agents were documented as potentially originating from multiple countries (Supplementary Table 8). In addition, we report the emergence of eight zoonotic diseases in Austria between 1975 and 2022, corresponding to a frequency of approximately one emerging zoonotic disease every six years. Notably, all of them were found to emerge in the past 20 years. The etiologic agents and their respective hosts, along with the year of discovery were: USUV (birds, 2001),

*Rickettsia helvetica* (*Ixodes ricinus*, 2005), *Anisakis* (human, 2009), *Brucella canis* (dog, 2010), *Rickettsia conorii* subsp. *raoultii* (dog, 2015), WNV (horse, 2016), *Thelazia capillipaeda* (domestic cat, 2018), and *Baylisascaris procyonis* (racoon, 2019). We documented three types of emergences: the first discovery outside the historical geographic range, the first discovery in Austria, and the first autochthonous case (Supplementary Fig. 9). Using the zoonotic web, we additionally estimated the current source range of the eight emerging zoonotic agents, revealing associations with 59 vertebrate hosts, including human, and four genera of arthropod vectors (Supplementary Fig. 10).

## Discussion

Cross-species transmission and emergence of zoonotic-origin diseases occur at complex animal-human-environment interfaces, within

dynamic social-ecological systems influenced by human behaviour, demographic shifts, and global changes. These interfaces represent significant One Health challenges. Here, we present the first attempt to analyse nearly 50 years of data on naturally occurring zoonotic infections (or contaminations) in Austria, leveraging an original One Health approach based on network theory. With approximately 80% of detections in animals and all those in vectors, food, and environmental matrices supported by direct evidence of zoonotic agents, we are confident in the robustness of our results. This work demonstrates that most zoonotic agents are capable of infecting both human and diverse animal species across various taxa, while evolving within multi-source, multi-agent ecological communities, consistent with the established principles in parasite community ecology[35]. We argue that the analysis of the zoonotic web holds greater value when studying potential zoonotic transmission chains compared to the commonly employed host-pathogen network approach, as it offers a broader epidemiological perspective and more analytical flexibility. Notably, we studied the centrality of zoonotic sources, including hosts, vectors, foodstuffs, and environmental matrices, within the network of zoonotic agent sharing, and evidenced that certain sources play a disproportionate role in the sharing of zoonotic agents. Specifically, we underscored the crucial role of arthropod vectors and foodstuffs (typically omitted in host-pathogen networks) in the risk of zoonotic disease emergence and transmission through the zoonotic web, pinpointing potential targets for One Health surveillance programmes.

Ten genera of zoonotic agents constituted 41% of the published research on zoonotic diseases in Austria, with seven of them involving agents subjected to compulsory surveillance and reporting in humans and/or animals[36]. This outcome underscores an imbalance in research interest, likely influenced by funding opportunities as well as global- and national-level prioritisation, typically based on known incidence and potential impact on human populations. Notably, diseases under European regulatory surveillance, such as those responsible for foodborne outbreaks or those posing a threat to global public health, like the influenza A virus, tend to receive more attention. Such a bias may lead to a skewed assessment of the overall zoonotic risk, especially concerning potentially "neglected" zoonoses such as certain helminth infections (e.g., dirofilariasis, dicrocoeliosis, hepatic capillariasis)[37]. Moreover, research trends show that very few publications in Austria address the environmental compartment, aligning with global observations[38].

From 1975 to 2022, Austria saw the emergence of eight zoonotic agents, averaging one EID every six years. While there is often an emphasis on viral emergence, particularly considering that RNA viruses pose the most significant threat[39], our findings offer a different perspective. Within our dataset, six out of eight emerging pathogens in Austria were bacteria and helminths. Notably, two of the emerging bacteria belong to the genus *Rickettsia*, aligning with the findings of Jones, et al.[4] This highlights the importance of broadening our focus beyond viral threats and acknowledging the substantial role that bacterial and helminthic pathogens play in the landscape of emerging diseases. Moreover, four emerging zoonoses are transmitted by arthropod vectors (WNV, USUV, *R. helvetica*, *R. conorii* subsp. *raoultii*). As a result of climate change and globalisation, there is a growing likelihood of new arthropod species populations becoming established in Austria, increasing the risk of future EID events[40]. Surprisingly, despite SARS-CoV-2 being notifiable for both humans and animals[41,42], none of the COVID-19-related publications concerning human cases refer to it as a zoonotic disease. Likewise, the sole publication investigating SARS-CoV-2 in Austrian animals did not mention its zoonotic potential[43].

Within the zoonotic web, multiple zoonotic sources contribute to the maintenance and spread of zoonotic agents. Studying the source-source network of zoonotic agent sharing is necessary to reveal indirect interactions[32], where one source influences another through shared agents. For example, if an agent is found in two sources, its prevalence in one may affect the other. However, these indirect interactions may lack epidemiological significance if, for instance, immunological or physical barriers prevent agent transfer between sources, such as when the sources do not share similar ecological niches[20]. Besides, many sources found (sero)positive for a zoonotic agent, may not, when taken individually, be able to maintain a sustained persistence of the agent within the network[44]. Nevertheless, as members of a zoonotic source community, interacting with maintenance and non-maintenance sources, they potentially play a role in the zoonotic agent ecology[45].

We observe that the zoonotic agent sharing network in Austria is organised into six communities. Our results indicate that the community, including humans, the oldest domesticated species (e.g., dog, cat, sheep, cattle, pig[46]), and synanthropic species (e.g., Norway rat, house mouse) share the most zoonotic agents. This suggests that the highest risk of zoonotic spillover originates from sources within this community. These national-level findings align with results from global studies[2,47]. In addition, human-modified environments, such as sandboxes, cluster with humans, domesticated and commensal species, highlighting the role of the shared ecosystem and environmentally persistent stages in the ecology of certain zoonoses[48]. The determinants of the zoonotic source community assembly and composition remain a challenge in disease ecology[5,49]. We found evidence that a limited number of highly connected zoonotic agents in the bipartite zoonotic web, such as USUV, *S. enterica*, WNV, and Influenza A, may, at least partly, drive zoonotic agent sharing community assemblage. The grouping of most food products into one community, predominantly sharing zoonotic agents typically associated with foodborne infections[50,51] (21/24 agents, 87.5%, including the five leading causes of foodborne diseases in the EU: *Campylobacter, Salmonella, Yersinia*, *E. coli* and *Listeria*[50], as well as 10 serovars of *Salmonella enterica* subsp. *enterica*), suggests that anthropogenic activities, particularly those related to food processing and transformation[52,53], may further influence the pattern of assembly within zoonotic source communities. These findings suggest that a combination of local epidemiological, ecological, human-related, and behavioural (e.g., relationship and proximity to human)[2] factors play a key role in shaping zoonotic agent sharing community patterns.

Our findings underscore the presence of central zoonotic sources in the network, demonstrating robust results across three to four centrality metrics after controlling for the research effort. These central zoonotic sources have a higher number of interactions with zoonotic agents, acting as hubs, or bridge different zoonotic source communities in the network, acting as connectors[54]. In particular, some livestock species (e.g., cattle, chicken), companion animals (e.g., dogs, cats, turtles), wildlife (e.g., yellow-necked field mouse, wild boar), and vectors (*Ixodes*) play a crucial role as bridge hosts, through which zoonotic agents can potentially spillover from maintenance (generally wild) host populations or communities to target populations (generally domesticated species or humans) that are usually "protected" through public health or biosecurity measures[25,44,55,56]. Notably, *Ixodes* ticks are pivotal in the epidemiology and zoonotic spillover of bacteria from the genera *Rickettsia, Borrelia*, and *Babesia*. Furthermore, the two communities involving USUV and WNV hosts illustrate the maintenance of zoonotic viruses within partially overlapping host communities. In this subsystem, mosquitoes of the genus *Culex* play a central role, serving as primary amplification vectors for WNV and USUV within each bird community. In addition, *Culex* mosquitoes act as bridge vectors between both avian maintenance communities and between these communities and potential mammalian hosts, including humans[57]. These results emphasise the importance of both vector monitoring and testing for pathogens as essential components for the early detection of emerging zoonoses and the establishment of early warning systems.

We present a novel approach based on the identification and quantitative characterisation of specific network structures, named One Health 3-cliques, for estimating the likelihood of zoonotic spillover at human-animal-environment interfaces. This method is flexible and can be applied to any zoonotic web. Our findings demonstrate that there is an increased co-occurrence of zoonotic agents at human-cattle and human-food interfaces, suggesting an elevated likelihood of zoonotic spillover. Notably, human zoonotic infection through consumption of contaminated food is a major public health risk, with *Listeria, Salmonella*, and *Escherichia* being the most frequently reported agents in food products across the included publications. Our results further emphasise the critical importance of monitoring zoonotic agents in food-processing environments.

A crucial challenge in formulating One Health surveillance and primary prevention strategies (i.e., at source)[17] for multi-source zoonotic agents, in particular emerging ones, is identifying what is the reservoir of infection[55], i.e., characterising, within a given context, the *"ecologic system in which an infectious agent survives indefinitely"*[58] and from which it can be sustainably transmitted to the target population[44]. The goal is to define what could be an optimal (high specificity and sensitivity) sentinel[59] to detect the circulation of a specific zoonotic agent above an acceptable threshold posing a potential transmission risk to the target population (typically human). Identifying sentinels through network metrics should depend on the topology of the network, the infectious agent to be monitored (e.g., endemic versus emerging, transmission route(s)), the (estimated) infection rate, the target population, the objective of the surveillance (e.g., early detection versus prevalence estimation)[60,61], and the specific epidemiological, ecological, and socio-cultural-economic context (e.g., what resources are available, what measures are acceptable, what is the community perception of the disease[62]). Selecting sentinels that are distant from each other in the network proved to enhance the overall probability of one sentinel being in proximity to an outbreak, thereby increasing the likelihood of detection[63]. For example, distributing the sentinels in different network communities[61] and prioritising surveillance of highly connected nodes in the network[30] (e.g., via regular sampling) would achieve higher performance than randomly selected nodes.

Nodes to be prioritised for surveillance may be different than those used for disease control[63]. Removing central nodes in the network, e.g., via vaccination or culling targeting "bridge" zoonotic sources (i.e., with high betweenness), can significantly reduce the connectivity of the zoonotic web[30], therefore decreasing the likelihood of zoonotic spillover into the human population. However, betweenness centrality fails to discriminate between zoonotic sources that have high betweenness because they have a lot of connections in the network, such as human and cattle, or sources that really connect two communities, serving as bottlenecks for zoonotic transmission flow[30] (e.g., *Ixodes*). Nevertheless, the effectiveness of interventions is intricately connected to the specific system under study and must be tailored to the context. For example, badger culling, equivalent to removing the *badger* node in the zoonotic web, has shown contrasting results on the prevalence of tuberculosis in cattle in the UK[64,65].

Alternative methodologies have been employed to investigate spillover events. For instance, Grange, et al.[66] ranked the spillover risk from known and newly discovered wildlife-origin viruses using a database of wildlife host-virus associations combined with expert opinion on drivers of spillover. Washburne, et al.[67] utilised percolation models to analyse cross-species transmission, uncovering inherent nonlinearity in spillover rate. In addition, Olival, et al.[8] used a dataset of mammal host-virus associations as a proxy for measuring spillover; using generalised additive models (GAMs), they identified predictors of host viral richness and estimated the number of undiscovered

viruses for each mammal species. Missing or unobserved links and nodes frequently occur in collected network data[68], which can impact the network properties. Diverse methods have been proposed to infer missing links[69–71] and nodes[69,72]. Notably, edge prediction accuracy can be enhanced through the use of network community structure[73]. These methods offer valuable mathematical and statistical approaches for future investigations of the zoonotic web, potentially allowing inference of zoonotic agent presence in a source where data has been lacking.

Our study acknowledges several limitations. First, poorly described taxonomic names hinder the precise identification of zoonotic agents or vertebrate hosts at the species level. Likewise, the unspecific description of food origin (e.g., "unspecified" animal), alongside our conservative approach to data validation/cleaning and adherence to authors' terminology, may have resulted in an inaccurate assessment of the degree centrality for some nodes. For example, Shiga toxin-producing *E. coli* (STEC) strains could refer to both VTEC and EHEC[74]; similarly, in the case of a host linked to both *Listeria* and *L. monocytogenes*, *Listeria* could potentially be *L. monocytogenes*. Imprecise description of the samples and zoonotic agents in publications represents a major limitation to the estimation of the zoonotic risk. Moreover, the single species-single pathogen approach, especially dominant in human medicine[11], and the tendency to disproportionally investigate zoonotic sources that are closer to humans can result in sample bias and in a skewed distribution of the number of zoonotic agents recorded per source, with human showing the highest number of zoonotic agents, followed by domesticated species. Zoonotic agent detection through environmental sampling remains scarce, potentially limiting result interpretation, particularly in a One Health context. Future zoonotic research could leverage environmental DNA/RNA (eDNA/ eRNA) sequencing for the detection and monitoring of zoonotic agents in the environment[75], as exemplified by SARS-CoV-2 wastewater-based surveillance[76]. Furthermore, despite efforts to control for research bias, our analysis is inevitably constrained by the existence of zoonotic source-agent associations that are either unknown or not yet published. This constitutes a major challenge in our understanding of zoonotic interactions. Ultimately, broadening the dataset by including additional data on natural infections documented in diverse laboratories (e.g., university laboratories that often investigate a broader range of sources and agents compared to national reference labs) as well as event-based surveillance (EBS) data sourced from various, non-official channels[77], could significantly enrich the dataset and enhance the depth of the analysis. ProMed-mail reports, for instance, benefit from evaluation by a multidisciplinary team of experts to ensure information reliability and accuracy before publication[78]. Future extensions could also explore conducting targeted searches for each zoonotic agent, based on available global lists[8,22–24], in the literature and international health organisation websites (e.g., World Health Organization, World Organisation for Animal Health). However, the latter may necessitate considerable time and resources; employing automated data extraction methods and tools could improve efficiency[79–81]. In addition, incorporating a temporal dimension to zoonotic source-agent interactions would allow for a more dynamic assessment of the zoonotic transmission chain within and between the communities. This approach could unveil seasonal variations in spillover events[82] as well as mechanisms that link host diversity to disease spread and emergence[83]. Moreover, as data on directionality in transmission is largely unavailable, we used a non-directed network and assumed a symmetrical process in interspecies transmission. This simplification may have limitations in capturing nuances in the dynamics of zoonotic transmission[84] (e.g., WNV can be transmitted from birds to humans via mosquitoes but this transmission process is not reciprocal). Furthermore, our data provides information on infection solely at the species level, overlooking individual

variations in shedding, and potentially missing key individuals acting as hubs ("superspreaders"). Finally, controlling for detection method stringency[8], such as PCR (or other direct detection methods) versus serology, could further refine our findings, allowing us to adjust edge weight within the network.

Here, we show that network analysis represents a cross-disciplinary method for unveiling the intricate web of zoonotic interactions involving multiple sources and infectious agents within an ecological system. In addition to presenting interactions between nodes, a zoonotic web approach enables the identification of influential zoonotic agents and sources that may hold epidemiological significance. Applying this approach across different settings, especially in regions identified as hotspots for zoonotic disease emergence, can expose critical knowledge gaps and reveal how existing epidemiological understanding, shaped by research data availability and funding priorities, may not always reflect on-the-ground realities. Overall, this work emphasises the need for further modelling and empirical studies to explore how maintenance is influenced by multiple source-agent interactions. Establishing efficient and context-adapted One Health network-based surveillance and control strategies requires supplementing the network analysis with multi-source data, ensuring a holistic, multidimensional understanding of the zoonotic web to unravel the complex dynamics of zoonotic transmission chains.

## Methods

### Systematic literature search and data extraction

The systematic literature search was conducted and reported according to the Preferred Reporting Items for Systematic Review and Meta-Analysis (PRISMA) guidelines[85].

Information about zoonotic agents circulating in Austria is dispersed across scientific papers, reports from the Austrian Agency for Health and Food Safety (AGES), reports from the Federal State Veterinary Services, and student theses. Between 17 July and 23 August 2022, a systematic literature search was conducted using the query ("Zoono*" AND ("Austria" OR "Österreich")) in the following databases: PubMed®, Scopus, and vetmed:seeker (internal database of the University of Veterinary Medicine Vienna, Austria), including articles published between the inception of the databases and the date of the search. Furthermore, the publication database of the AGES was searched using the keyword "zoono". Additional papers found in the reference section of reviews that provided relevant information were also included. Retrieved publications were deduplicated in the reference manager Citavi (Swiss Academic Software. 2023) before the following selection processes.

Titles and abstracts were first screened for relevance using the following inclusion criteria: the publication presented data pertaining to at least one zoonotic disease or agent that was investigated or documented in Austria, and the agent was identified as zoonotic in the paper. Publications were excluded (i) if they did not investigate or describe a zoonotic disease that was identified as such, (ii) if research was not conducted in Austria, (iii) if publications did not describe naturally occurring zoonotic infection, or (iv) if publications described disease physiology or (v) dealt with treatment or methods for pathogen detection. Book chapters, posters, literature reviews, statistical forecasts, and conference proceedings were excluded. Regarding antimicrobial resistant bacteria, papers were included if they specifically explored the animal-human interface and/or the authors referred to zoonotic transmission. To prevent duplication of data, diploma-, master's-, and doctorate thesis were not included if a peer-reviewed research paper published the same data.

In a second step, the full texts of the previously selected titles/abstracts were screened using the inclusion/exclusion criteria described above. Publications were excluded if they were not in German or English language or did not describe the situation in Austria. When a publication dealt with multiple countries, it was included if it provided specific information on zoonotic diseases in Austria.

The following data was extracted from the selected publications: **(i) Publication data**: citation, year of publication, and type of publication; **(ii) Type of study**: case study, original research, or national surveillance data; **(iii) Investigated zoonotic agent**: agent type (e.g., bacterium, virus, parasite, fungus, prion, or other) and common/scientific names as mentioned in the information source; **(iv) Investigated host**: host category, e.g., human, companion animal (defined as domesticated animals possessed by a person for reasons other than food or resource production, including domesticated small rodents or exotic companion animals), livestock (defined as domesticated animals kept for resource and food production), wildlife (defined as free-ranging or captive wild animal species that are not domesticated), common/scientific names as mentioned in the information source, if the zoonotic agent was detected in the host, i.e., seropositive (confirmed by the presence of antibodies), positive (direct detection of the agent), or negative; **(v) Investigated vector:** common/scientific names as mentioned in the information source, and if the zoonotic agent was detected in the vector (positive/negative); **(vi) Investigated environmental matrix** and if the zoonotic agent was detected in the matrix (positive/negative); **(vii) Investigated food matrix**: the specific type of foodstuff investigated, the origin of the food product (animal or plant), and if the zoonotic agent was detected in the foodstuff (positive/negative); **(viii) Epidemiological context:** study year, federal state(s), whether the case was imported and most probable origin, whether the zoonotic agent was mentioned as emerging in Austria, and whether specific professional activities were deemed to carry an elevated risk of exposure.

### Data curation

First, the data underwent quality control and cleaning procedures where the unique values of each field were checked to search for inaccurate or missing data in the dataset using the R function *unique()*. Events containing detected errors were manually inspected against the original data source, and, when necessary, the erroneous values were modified, replaced, or removed. Furthermore, for each animal host, vector, and zoonotic agent, common and scientific names, as well as taxonomic classification were resolved against the NCBI Taxonomy database[86] using the R package taxize[87]. If a conflict occurred between the scientific name and/or common name as provided in the information source and the NCBI-resolved name, the information source was cross-referenced and searched for complementary information on the investigated species. When the original source did not provide sufficient details for the identification of a scientific name, the most precise taxonomic denomination was used.

Food categories were generated by combining the food source (e.g., cattle) and the type of food (e.g., meat and meat products). For analytical purposes, foodstuffs designating the same type of food were grouped. For example, "kebab", "ground meat", or "rillettes" were coded as "meat and meat product"; "milk" and "milk product" were coded as "dairy"; "egg" and "egg product" were coded as "egg"; "fish", "fish filet", or "rollmops" were coded as "marine product"; "salad", "spices", "fruit", or "vegetable" were coded as "plant-based food". The categories "cheese" (e.g., mozzarella, Brie, Roquefort) and "sausage" (e.g., ham, salami, raw meat sausage) were also added for more accurate representation.

### Analysis of the zoonotic web

The dataset was used to create an undirected network representing the web of naturally occurring zoonotic interactions, thereafter called the *zoonotic web*, depicting the relationships between zoonotic actors. In this network, the zoonotic agents and their zoonotic sources (i.e., vertebrate hosts, arthropod vectors, foodstuffs, and environment)

were shown as nodes linked by edges, which represented zoonotic infection (hosts and vectors) or colonisation (food and environmental sources). A link between a zoonotic agent *i* and a vertebrate host *j* indicates that agent *i* was directly or indirectly detected in host *j*. A link between a zoonotic agent *i* and a vector *j* signifies that agent *i* was identified in vector *j*, implying that vector *j* may transmit agent *i* to a vertebrate host through a bite or mechanically. Likewise, a link between a zoonotic agent *i* and an environmental matrix *j* indicates the presence of agent *i* in environment *j*, potentially leading to infection of a vertebrate host upon contact. Lastly, a link between a zoonotic agent *i* and a food matrix *j* indicates that agent *i* was detected in food *j*, which may result in the infection of a vertebrate host through ingestion. In the network, the most specific NCBI-resolved zoonotic agent and host names were employed while arthropod vectors were aggregated at the genus level. The zoonotic web is a bipartite network, i.e., a graph that contains two disjoint sets of nodes, the zoonotic sources and the zoonotic agents, respectively, such that every edge connects the two node sets (i.e., interactions among zoonotic sources or among zoonotic agents were not allowed). The degree centrality (the number of links a node has) was calculated for each node. In the epidemiological context, the node degree centrality for each zoonotic source corresponds to the zoonotic agent richness, i.e., the number of taxa directly or indirectly detected in the zoonotic source. Similarly, the node degree centrality for each zoonotic agent corresponds to the zoonotic source range, i.e., the number of sources from which it has been directly or indirectly evidenced, reflecting its "host" or "zoonotic source" plasticity.

The zoonotic source-agent network was subsequently projected into a one-mode network of zoonotic agent sharing among sources. Edges were weighted by the number of shared zoonotic agents between two sources. By transforming the zoonotic source-agent bipartite network into a source-source unipartite network, a "transmission-potential network"[29] was created, where sources were linked based on shared zoonotic agents. To account for research biases, we considered, for each source, the total number of zoonotic investigations (i.e., the number of times a source was studied). For instance, if, in one study, a source was investigated annually for three years, we counted three zoonotic investigations. Similarly, if the same source was examined for five zoonotic agents in a single study, we counted it as five zoonotic investigations. This approach provided a more accurate estimation of scientific research effort compared to simply counting the number of studies. We used the number of zoonotic investigations as an estimate of scientific research effort for each source and regressed each edge weight by the Box-Cox transformed number of zoonotic investigations of the least studied source of each edge. The residuals were subsequently rescaled so that the lowest weight value was 1[9,10]. After removal of the isolated components in the research effort-adjusted one-mode network of zoonotic agent sharing, we calculated the following node centrality metrics: degree centrality, strength centrality (the sum of the weights of edges to/from a node), weighted betweenness (the number of shortest paths that go through a node, which allows identifying nodes that act as bridges connecting the different communities), and weighted closeness (the average inverse distance to all other nodes)[31]. To calculate the weighted metrics, the edge weight was transformed into cost by dividing 1 by the weight[88]. Node rankings through node centrality metrics were compared using the Kendall correlation test. Average values of the node centrality metrics were also compared between the four zoonotic source categories using the Kruskal-Wallis test. When a difference was evidenced, a pairwise comparison between zoonotic source categories was performed using the Wilcoxon rank sum test; *p*-values were adjusted following the Benjamini-Hochberg method[89]. Network analyses were performed using the R packages igraph[90] and bipartite[91].

## Community detection

We used the Leiden algorithm[92], which relies on a measure called modularity[93], to detect communities of zoonotic agent sharing within the research-adjusted one-mode network of zoonotic sources. The method aims to optimise modularity by maximising the difference between the actual and expected number of edges within communities. The Leiden algorithm is considered as an improvement over the Louvain algorithm[94]. It comprises three distinct steps: initial optimisation of modularity, subsequent refinement of the partition, and a third step focusing on the community aggregation process[92]. Notably, by refining the local partition in each community, the Leiden algorithm demonstrates enhanced stability in community detection and offers more efficient computation time compared to the Louvain algorithm[94].

## Exploring network One Health cliques

We investigated the circulation of zoonotic agents at human-animal-environment interfaces within the research effort-adjusted network of zoonotic agent sharing by searching "One Health" 3-cliques in the network structure. A clique is a fully connected subgraph within the network. A 3-clique is a set of three nodes all pair-wisely connected to each other, therefore forming a triangle[31]. We were interested in 3-cliques that included nodes representing the three traditional One Health compartments, i.e., animal (173 nodes), human (one node), and environment (15 nodes). Plant-based foodstuffs, invertebrate vectors, and any environmental matrices (including surfaces and tools in food processing plants) were included into the compartment "environment" while food products of animal origin were considered within the "animal" compartment. We ranked the One Health cliques by their total edge weight, i.e., the sum of the edge weights between all pairs of nodes (zoonotic sources) within the clique, corresponding to the research effort-adjusted sum of zoonotic agents shared between all pairs of One Health compartments.

## Reporting summary

Further information on research design is available in the Nature Portfolio Reporting Summary linked to this article.

## Data availability

The raw dataset generated in this study, as well as its cleaned and validated version, are available in the Supplementary Code. The data has also been archived in the study repository on figshare with the identifier: https://doi.org/10.6084/m9.figshare.25306177[95].

## Code availability

All analyses were conducted in R Statistical Software version 4.3.0 (2023-04-21) "Already Tomorrow". The documented R scripts used for data cleaning, validation, processing, and analysis, are available in Supplementary Code. The READ.ME file contains the necessary instructions to run the code and replicate our results. These files have also been archived in the study repository on figshare[95].

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

## Acknowledgements

The authors thank Sina Sajjadi and Rafael Prieto-Curiel for their valuable insights and advice on network analysis.

## Author contributions

Initial concept: A.D-L.; Analysis design: A.D-L.; Data collection: A.V. and A.D-L.; Data collation and processing: A.D-L. and A.V.; Data analysis and modelling: A.D-L. and G.A.P.; Visualisation: A.D-L., L.Y. and G.A.P.; Writing (initial draft): A.D-L. and A.V.; Writing (review and editing): A.D-L., A.V., G.A.P., L.Y., A.J. and A.K.

## Competing interests

The authors declare no competing interests.
