## [Peer Review File · Nature Communications]

A One Health framework for exploring zoonotic interactions demonstrated through a case studyREVIEWER COMMENTS

Reviewer #1 (Remarks to the Author):

General comments

I want to thank the author for their paper "A One Health framework for exploring the zoonotic web: a case study" This comprehensive analysis of published data on zoonotic interactions in Austria over a nearly 50-year period is of interest for the scientific community working on zoonosis and on the complex interactions between animals, human, vector and environment leading to disease emergence and transmission.

- This network-based approach, if applied elsewhere could help designing relevant One Health strategies against zoonoses.
- The methodology is original, data analysis are well performed and the figures are very informative and well presented.
- The main original results include overview of zoonotic events in Austria for a 50 year period and the zoonotic web methodology allow to understand interactions leading to zoonotic events
- The network of zoonotic agent sharing approach (and the analysis of centrality or high degree nodes) is interesting to understand the potential role of different species in the contact with others, in the transmission or as reservoirs. Therefore, it helps in the understanding of potential spill over.
- The analysis identifies 6 host communities with specific zoonotic agent diversity.
- The analysis also allows to review importation cases of zoonotic agents (48) and emergence (8) in Austria.
- Emerging zoonotic agents where associated to different sources of vertebrate hosts or of arthropods, this is the first use of the zoonotic web approach (see Imported and emerging zoonotic agents in the results)
- I like the practical propositions regarding optimal sentinels to detect the circulation of a specific zoonotic agent (discussion section 342-353)

My main concerns

In the discussion, the authors argue that the comprehensive analysis of the zoonotic web holds greater value compared to the commonly employed host pathogen network approach (260-262), especially since they have included environmental sources. However, they include vectors in these environmental sources and vectors are also often included in host-vector-pathogens network approach. Therefore, it limits the originality of the paper to the inclusion of the foodstuffs and environmental matrices, which in practical represent a small percentage of data in their dataset. It is not their fault because these data are scarce but it limits the interpretation of some of their results. And I would have appreciated a discussion regarding this limit and their idea to improve in the future the situation for example through environmental studies using eDNA for example. Another limit of the study is the data source used and their origin (published papers) which does not include reports. This is partially discussed indicating that datasets from diverse laboratories could be added) (I377) but the authors do not discuss the possible inclusion of international organizations (WHO and WOA) or European dataset (ECDC, EFSA) and of informal data that can be obtained by text mining on the web. (Dub et al. BMC Public Health (2023) 23:1488 <https://doi.org/10.1186/s12889-023-16396-y>)

I am also not convinced by the One Health 3 cliques approach since the dataset lack environmental data, therefore, since the authors search for interactions between the 3 sectors, it seems that the same sources are always present in the 3 cliques approaches. I am also surprised not to see in these results the classical triangle host-vector pathogen.

Some discussion is lacking especially around other methodologies to analyses spill over events, and also on how to use this methodology to identify not yet known vectors or hosts.

Specific comments

Title

I understand the idea to have a fancy title however I found the title not very informative since the zoonotic web is a proposed new concept. I would suggest to go for a title more informative like "a

comprehensive analysis of published data on zoonotic interactions in Austria. New tool to improve understanding of transmission and surveillance. However, the editor is the best person to give and advice on this

Introduction

38-39 The paper from Jones indicates that EID events are dominated by zoonoses (60.3% of EIDs): the majority of these (71.8%) originate in wildlife, it does not mean that "more than 70% of EID are caused by pathogens with a wildlife origin, it is 71.8% of 60%

62-63: I would add "human behavior" and not only "proximity to the animal species"

68-69 "monitor and mitigate potential spillover event" I think we can monitor indicators that can lead to a spillover and effective circulation in different species, more than monitoring the spillover
74-75 Without comparisons with other countries, it is not obvious with these figure that Austria is especially a hot spot of biodiversity or a place where interactions between human or wildlife is specifically high.

76 Austria adheres to a combination of European and national regulations: which ones? One Health regulations or specific regulation on human health and animal health. What about regulations from WHO and WOAHA?

80 why are the collected data "newly"

Results

- Dataset

94 "were not eligible" need to be a bit more specific (even if it is described in the mat met section)

102-108 the increase level of publications on zoonosis in Austria or worldwide is not informative itself if you do not compare to the general increase of publications between the two period (for example publications with health as a ke word increased by x7 between the 2 periods)

- Research trends

123-125 It would be interesting to separate vectors from environment and food to describe the trends to see if the most gradual increase is due to publication in environmental matrix or in vectors

- Zoonotic web actors

132-134 an additional comparative analysis of zoonotic richness between human, non-human mammals and birds would be useful

- Imported and emerging zoonotic agents

243 "emerged in the past 20 years", I would say "where found to emerge" since this is related to existing publications and depends of the dataset

Discussion

254 : I would add "including human behavior and demographic and global changes"

257 why "novel" dataset for a bibliographical dataset

280 coronavirus are not in the dataset of emerging zoonotic agents, it is discussed later in the paper but it emphasize the fact that the dataset selection using "zoono" as a criteria has some limits since papers can described emergence without mentioning zoonosis in the text. Other emergences could have been missed, therefore it is difficult to conclude on "most emerging pathogens in Austria are Bacteria and helminths"

293 many sources (sero) positive, It would be nice to have in the results and discussion (and not only in the methodology section) some explanation and analysis about the papers found to described either seropositive animals/humans or with pathogen identification. It has some impact on robustness of the results regarding types of emergence and on surveillance strategies.

296 replace "demonstrated" by "observed"

306 Can you really conclude on the driving forces of anthropogenic activities with the grouping of most food products into one community. Do you have enough data on food product to conclude?

Reviewer #2 (Remarks to the Author):

The authors of this study investigated the network of hosts sharing microbiological agents identified as zoonotic in literature published in Austria from 1975 to 2022. While the study reveals

the complexity of a microbiological agent-sharing network amongst various hosts based on the vast datasets, the manuscript require revision with particular attention to the following points.

First, in the zoonotic web the authors constructed, edges represent the sharing of microbiological agents between hosts. For these agents and hosts, "the most specific NCBI-resolved zoonotic agent and host names" are selected and matched during network construction. However, the articles that formed the basis of this network construction were selected only when they included "Zoono*". I wonder if this process would have removed articles whose information could form edges according to the process of network construction but were removed because they simply did not define the agents studied as zoonotic. Would the network constructed based on literature search using a list of zoonotic microbiological agents, instead of using "Zoono*", be different from the network presented in the manuscript?

Second, the authors attempted to reduce bias from research efforts by accounting for the number of studies for the same sources and agents. While this might address the quantity of investigations, would this also address the presence or absence of investigations on particular sources or agents? Almost every network property seems centered around humans, followed by livestock and companion animals. Does the approach used in this study address research efforts not conducted ever due to a lack of interest? (although I also acknowledge that this could indicate a potential absence of zoonotic agents and thus no edge). The manuscript requires an in-depth discussion of potential limitations in their network construction processes.

There needs to be clearer explanations or discussion on the epidemiological meaning of individual network features. For example, some of the indirect connections in the giant connected network component might not have epidemiological implications if the transfer of zoonotic agents is not possible due to immunological barriers.

The following are specific comments:

Line 81: The full web can be misleading because the network construction is based on literature review, which has some limitations as mentioned above.

Line 93: Although explained in the method section, it would be good to mention briefly the screening terms used for literature review here.

Line 136: If USUV is not mentioned earlier, please write the full name here.

Line 149: I would use Figure S6 in the main text (with/without Figure 3). Figure S6 is more intuitive. In Figure S6, please highlight edges between major sources and agents (e.g., edges between the top 10 sources and agents) in bold lines.

Lines 148 onwards: Define network metrics briefly, although they are defined in the method, to help readers without knowledge of network analysis.

Line 181: "Research effort-adjusted" is too vague and might give the impression that there is no bias unadjusted. Briefly explain how this is done when this is first mentioned in the manuscript.

Line 187: List and briefly define the list of four centrality metrics.

Line 200: What is the difference in epidemiological meaning between closeness and degree centrality metrics?

Reviewer #3 (Remarks to the Author):

This paper described the analysis of zoonotic pathogen spillovers between human-animal and environment in Austria, based on published and grey literature. Understanding the spillover mechanisms and links with human practices is key to develop target and risk based surveillance and to prevent zoonotic disease emergence at source. Moreover the work is original by the method used as the authors have used network analysis and One-Health 3 cliques to characterise the spillover events.

This work present important and significant results, highlighting the need to assess the real figures of zoonotic agents circulating in the territory, showing the recent consideration of the environment component in disease transmission research, showing that beyond viral threats, disease emergence caused by bacteria has been overlooked.

The work is also strong as it highlights its own limits, mentioning that the results might be skewed towards human, and domesticated animals as they have been the focus of research, compared to other species and pathogens less studies. Nevertheless the result remains relevant and interesting, the application of such an approach into other setting e.g. hot spot developing countries would be very valuable to highlights similar gaps in understanding but also to demonstrate how our epidemiological knowledge which is used for health strategy decision making is highly skewed based on research data available (and therefore funding priorities) which do not always reflect the field reality. A mention to this potential application in the conclusion and perspective would further improve this publication.

Additional comments:

As the authors highlight very well that the data might be skewed to zoonotic sources closer to humans; it might be worthwhile to mention that the data might also be skewed towards zoonotic agents which have been priorities internationally as pandemic threats and therefore highly studied and under high active surveillance (e.g. Avian Influenza- poultry, Rabies-dogs, BSE-cattle). The authors mention page 9 and 10 from line 338 that one crucial element of surveillance to prevent emergence at community level is the use of sentinel - It is important to take into perspective that sentinels, as an active component of surveillance allows to generate additional knowledge and better understanding of pathogens circulation and spillover, might also allow to early detect an infection but only if the event based component of the community surveillance is effective, I.e. the communities are aware of risks and trained on how to recognize them and undertake quick action to prevent spillover or spread and alert relevant stakeholder to take action. This should be specified in the manuscript - this work requires also tailoring to the specific socio-economic context where the pathogen evolves, which ties in nicely with this work. Reference to the recent work of Guenin et al. PLOS NTD could be made.

Point-by-point response to the reviewers' comments for Desvars-Larrive et al.'s manuscript "A One Health framework for exploring zoonotic interactions: a case study"

We sincerely thank the reviewers for their valuable feedback and constructive comments on our manuscript. Their insights have greatly contributed to improving the clarity and comprehensiveness of our work. We have carefully addressed each point raised and incorporated necessary revisions to enhance the quality and impact of the manuscript.

1. Reviewer #1

(Remarks to the Author):

General comments

I want to thank the author for their paper "A One Health framework for exploring the zoonotic web: a case study" This comprehensive analysis of published data on zoonotic interactions in Austria over a nearly 50-year period is of interest for the scientific community working on zoonosis and on the complex interactions between animals, human, vector and environment leading to disease emergence and transmission.

- This network-based approach, if applied elsewhere could help designing relevant One Health strategies against zoonoses.
- The methodology is original, data analysis is well performed and the figures are very informative and well presented.
- The main original results include overview of zoonotic events in Austria for a 50-year period and the zoonotic web methodology allow to understand interactions leading to zoonotic events
- The network of zoonotic agent sharing approach (and the analysis of centrality or high degree nodes) is interesting to understand the potential role of different species in the contact with others, in the transmission or as reservoirs. Therefore, it helps in the understanding of potential spill over.
- The analysis identifies 6 host communities with specific zoonotic agent diversity.
- The analysis also allows to review importation cases of zoonotic agents (48) and emergence (8) in Austria.
- Emerging zoonotic agents were associated to different sources of vertebrate hosts or of arthropods, this is the first use of the zoonotic web approach (see Imported and emerging zoonotic agents in the results)
- I like the practical propositions regarding optimal sentinels to detect the circulation of a specific zoonotic agent (discussion section 342-353)

We greatly appreciate your positive feedback and the insightful suggestions you provided for our manuscript. Below, you will find a detailed response addressing each of your points.

My main concerns

- 1.1. In the discussion, the authors argue that the comprehensive analysis of the zoonotic web holds greater value compared to the commonly employed host pathogen network approach (260-262), especially since they have included environmental sources. However, they include vectors in these environmental sources and vectors are also often included in host-vector-pathogens network approach. Therefore, it limits the originality of the paper to the inclusion of the foodstuffs and environmental matrices, which in practical represent a small percentage of data in their dataset. It is not their fault because these data are scarce but it limits the interpretation of some of their results.

And I would have appreciated a discussion regarding this limit and their idea to improve in the future the situation for example through environmental studies using eDNA for example.

We thank Reviewer #1 for this insightful comment. In the paper, nodes representing vectors and those representing the environment are distinguished by colour in the network visualization (yellow for the former, green for the latter) and analysis (e.g., Table 2). To facilitate trend analysis and conduct the 3-cliques detection, addressing the One Health compartments, we have merged these two categories alongside plant-based food, into the compartment “environment” (this is specified in the Methods section as well as in the figure caption when relevant). As suggested, we have included a discussion regarding this limitation and propositions on how improving this issue in the future (Discussion → section on the study limitations):

Zoonotic agent detection through environmental sampling remains scarce, potentially limiting result interpretation, particularly in a One Health context. Future zoonotic research could leverage environmental DNA/RNA (eDNA/ eRNA) sequencing for the detection and monitoring of zoonotic agents in the environment (Bass et al., 2023), as exemplified by SARS-CoV-2 wastewater-based surveillance (Amman et al. 2022).

Supporting references:

Bass, D., Christison, K. W., Stentiford, G. D., Cook, L. S. J. & Hartikainen, H. Environmental DNA/RNA for pathogen and parasite detection, surveillance, and ecology. *Trends Parasitol.* 39, 285-304 (2023).

Amman, F. et al. Viral variant-resolved wastewater surveillance of SARS-CoV-2 at national scale. *Nature Biotechnology* 40, 1814-1822 (2022).

1.2. Another limit of the study is the data source used and their origin (published papers) which does not include reports. This is partially discussed indicating that datasets from diverse laboratories could be added (I377) but the authors do not discuss the possible inclusion of international organizations (WHO and WOAHA) or European dataset (ECDC, EFSA) and of informal data that can be obtained by text mining on the web. (Dub et al. *BMC Public Health* (2023) 23:1488 <https://doi.org/10.1186/s12889-023-16396-y>)

Actually, the dataset also incorporates reports from the Austrian Agency for Health and Food Safety (AGES), which serves as National Laboratory or Reference Center for several zoonotic diseases in Austria (<https://www.ages.at/en/ages/reference-centres-laboratories>). This inclusion is detailed in the Methods section: “Furthermore, the publication database of the AGES (<https://www.ages.at/en/research/research-portal>) was searched using the keyword “zoono”. Therefore, the dataset includes data extracted from the annual reports of the national laboratory, available at: <https://www.ages.at/en/human/disease/zoonoses#c18248>.

The presence of reports in the included literature was specified in the Results section (→ Dataset of zoonotic interactions in Austria, 1975-2022):

“(…) leading to a total of 246 publications that were ultimately included in this study (168 scientific articles, 13 reports, and 65 theses)”

To enhance clarity, we have revised the first sentence of the final paragraph in the introduction. Revised text:

In this study, we extracted data from scientific papers and national laboratory reports, spanning 47 years of publications, to generate a real-world network describing the full web of zoonotic interactions in Austria and characterise the various interfaces through which zoonotic spillover may occur.

We did not include nor discuss data from the ECDC and EFSA because national data on zoonotic disease occurrence and foodborne outbreaks are transmitted by the Austrian Agency for Health and Food Safety (AGES) to ECDC/EFSA, following the European regulatory framework, for the production for different reports, such as the joint “European Union One Health Zoonoses Report.” Consequently, including ECDC/EFSA data would lead to duplications of already integrated data (see above).

The authors are unaware of any World Health Organization (WHO) database or reports detailing zoonotic disease occurrence per country. While the WHO Disease Outbreak News (DONs) (<https://www.who.int/emergencies/disease-outbreak-news>) could be an interesting data source, the

search for “Austria” conducted on 9 April 2024 hit only three results. WOAH-WAHIS reports concern only WOAH-listed diseases in domestic animals and wildlife, as well as emerging and zoonotic diseases. However, these reports do not align with our inclusion criteria, which limit reports/publications to zoonotic agents identified as zoonotic by the authors. Nevertheless, we recognise the potential for future expansion of our work by incorporating data from this source.

As proposed by Reviewer #1, incorporating event-based surveillance (EBS) data sourced from diverse, non-official channels could significantly enrich the dataset (Dub et al., 2023). A notable example of this approach is the ProMed-mail reports (<https://promedmail.org/>), which present the advantage that a multidisciplinary team of subject matter expert moderators evaluates the reliability and accuracy of the information before publication (Yu & Maddoff, 2004). However, it's worth noting that posts older than 30 days are not freely accessible.

Therefore, future extensions of the dataset and analysis could explore i) incorporating ProMed-mail reports and ii) conducting a targeted literature search for zoonotic agents based on available global lists (e.g., Mollentze & Streicker, 2020; Olival et al., 2017, Carlson et al., 2022; Wardeh et al., 2015), i.e., not limited to the zoonotic agents identified as zoonotic by the authors. However, the latter point may require significant time investment, and utilising text mining tools for web search, as suggested by Reviewer #1, could enhance efficiency.

These points have been summarised in the Discussion (→ last paragraph describing the study limitations):

Ultimately, broadening the dataset by including additional data on natural infections documented in diverse laboratories (e.g., university laboratories that often investigate a broader range of sources and agents compared to national reference labs) as well as event-based surveillance (EBS) data sourced from various, non-official channels (Dub et al., 2023), could significantly enrich the dataset and enhance the depth of the analysis. ProMed-mail reports, for instance, benefit from evaluation by a multidisciplinary team of experts to ensure information reliability and accuracy before publication (Yu & Maddoff, 2004). Future extensions could also explore conducting targeted searches for each zoonotic agent, based on available global lists (Olival, K. J. et al, 2017; Carlson et al. 2022; Wardeh et al., 2015), in the literature and international health organisation websites (e.g., World Health Organization, World Organisation for Animal Health). However, the latter may necessitate considerable time and resources; employing automated data extraction methods and tools could improve efficiency (Gutiérrez-Sacristán, A. et al., 2017; Paraskevopoulos et al., 2022; Walker et al., 2022).

Supporting references:

Carlson, C. J. et al. The Global Virome in One Network (VIRION): an atlas of vertebrate-virus associations. *mBio* 13, e02985-02921 (2022).

Gutiérrez-Sacristán, A. et al. Text mining and expert curation to develop a database on psychiatric diseases and their genes. *Database* 2017 (2017).

Mollentze, N. & Streicker, D. G. Viral zoonotic risk is homogenous among taxonomic orders of mammalian and avian reservoir hosts. *Proc Natl Acad Sci U S A* 117, 9423-9430 (2020).

Olival, K. J. et al. Host and viral traits predict zoonotic spillover from mammals. *Nature* 546, 646-650 (2017)

Paraskevopoulos, S., Smeets, P., Tian, X. & Medema, G. Using Artificial Intelligence to extract information on pathogen characteristics from scientific publications. *Int. J. Hyg. Environ. Health* 245, 114018 (2022).

Walker, V. R. et al. Evaluation of a semi-automated data extraction tool for public health literature-based reviews: Dextr. *Environ. Int.* 159, 107025 (2022).

Wardeh, M., Risley, C., McIntyre, M. K., Setzkorn, C. & Baylis, M. Database of host-pathogen and related species interactions, and their global distribution. *Sci. Data* 2, 150049 (2015).

1.3. I am also not convinced by the One Health 3 cliques approach since the dataset lack environmental data, therefore, since the authors search for interactions between the 3 sectors, it seems that the same sources are always present in the 3 cliques approaches. I am also surprised not to see in these results the classical triangle host-vector pathogen.

We agree with Reviewer#1 regarding the dataset's limited environmental data. Nonetheless, within the One Health 3-clique approach (as in the trend analysis), we included three types of zoonotic sources in the environmental compartment, i.e., nodes representing plant-based food, vectors, and environmental matrices, totalling 15 nodes in our analysis. Indeed, “vector” is not mentioned in the traditional One

Health triad, which refers to human, animal, and environment. This was specified into the section Methods → Exploring network One Health cliques: “*Plant-based foodstuffs, invertebrate vectors, and any environmental matrices (including surfaces and tools in food processing plants) were included into the compartment “environment” while food products of animal origin were considered within the “animal” compartment.*” It is also acknowledged in the caption of Figure 6a.

Furthermore, for enhanced clarity, we have included the number of nodes within each One Health compartment of the zoonotic web (Methods → Exploring network One Health cliques):

We were interested in 3-cliques that included nodes representing the three traditional One Health compartments, i.e., animal (173 nodes), human (one node), and environment (15 nodes).

Despite potential limitations in our analysis due to the relatively small number of nodes representing the environmental compartment, we believe that the 3-clique approach holds value for other researchers, particularly those dealing with larger datasets. Indeed, few approaches in One Health currently incorporate a quantitative aspect to the animal-human-environment interface, making our method potentially valuable for advancing research in this area.

1.4. Some discussion is lacking especially around other methodologies to analyses spill over events, and also on how to use this methodology to identify not yet known vectors or hosts.

We thank Reviewer #1 for the insightful comment. In response, we have integrated a new paragraph into the Discussion section to acknowledge previous research and explore alternative methodologies applicable to our dataset. Considering the number of studies investigating spillover events and identifying unknown hosts/vectors of zoonotic agents, we have highlighted a couple of studies that exemplify diverse approaches:

Alternative methodologies have been employed to investigate spillover events. For instance, Grange, et al. ranked the spillover risk from known and newly discovered wildlife-origin viruses using a database of wildlife host-virus association combined with expert opinion on drivers of spillover. Washburne, et al. utilised percolation models to analyse cross-species transmission, uncovering inherent nonlinearity in spillover rate. Additionally, Olival, et al. used a dataset of mammal host-virus associations as a proxy for measuring spillover; using generalised additive models (GAMs), they identified predictors of host viral richness and estimated the number of undiscovered viruses for each mammal species. Missing or unobserved links and nodes frequently occur in collected network data (Lü et al., 2011), which can impact the network properties. Diverse methods have been proposed to infer missing links (Kim & Leskovec, 2011; Clauset et al., 2008; Guimerà & Sales-Pardo, 2009) and nodes (Kim & Leskovec, 2011; Liu & Zhou, 2011). Notably, edge prediction accuracy can be enhanced through the use of network community structure (Yan & Gregory, 2012). These methods offer valuable mathematical and statistical approaches for future investigations of the zoonotic web, potentially allowing inference of zoonotic agent presence in a source where data has been lacking.

Supporting references:

Clauset, A., Moore, C. & Newman, M. E. J. Hierarchical structure and the prediction of missing links in networks. *Nature* 453, 98-101 (2008).

Grange, Z. L. et al. Ranking the risk of animal-to-human spillover for newly discovered viruses. *PNAS* 118, e2002324118 (2021).

Guimerà, R. & Sales-Pardo, M. Missing and spurious interactions and the reconstruction of complex networks. *PNAS* 106, 22073-22078 (2009).

Kim, M. & Leskovec, J. in *Proceedings of the 2011 SIAM International Conference on Data Mining (SDM)*. 47-58.

Lü, L. & Zhou, T. Link prediction in complex networks: A survey. *Physica A: Statistical Mechanics and its Applications* 390, 1150-1170 (2011).

Olival, K. J. et al. Host and viral traits predict zoonotic spillover from mammals. *Nature* 546, 646-650 (2017).

Washburne, A. D. et al. Percolation models of pathogen spillover. *Philosophical Transactions of the Royal Society B: Biological Sciences* 374, 20180331 (2019).

Yan, B. & Gregory, S. Finding missing edges in networks based on their community structure. *Phys. Rev. E* 85, 056112 (2012).

Specific comments

Title

1.5. I understand the idea to have a fancy title however I found the title not very informative since the zoonotic web is a proposed new concept. I would suggest to go for a title more informative like “a comprehensive analysis of published data on zoonotic interactions in Austria. New tool to improve understanding of transmission and surveillance. However, the editor is the best person to give and advice on this

Thank you for your input on the title. We have adhered to Nature Communications' guideline of limiting the title to 15 words. In response to Reviewer #1's suggestion, we have changed "zoonotic web" to "zoonotic interactions" for clarity. While "zoonotic web" is explained later, we agree that a more informative title is preferable. We have retained "case study" to highlight the applicability of our research methodology beyond Austria. As we did not explore surveillance strategies (e.g., by developing surveillance scenarios and testing the use of some nodes in surveillance), we opted not to include "surveillance" in the title. We believe the revised title, "*A One Health framework for exploring zoonotic interactions: a case study*", meets the criteria and addresses the feedback provided. We hope this title is satisfactory to Reviewer #1 and the Editor.

We have accordingly changed title name on the different documents: Supplementary Information 1 (R scripts and READ.ME file), Supplementary Information 2, Supplementary Information 3, and figshare repository.

Introduction

1.6. 38-39 The paper from Jones indicates that EID events are dominated by zoonoses (60.3% of EIDs): the majority of these (71.8%) originate in wildlife, it does not mean that “more than 70% of EID are caused by pathogens with a wildlife origin, it is 71.8% of 60%

We thank Reviewer #1 for bringing this to our attention. That was indeed a mistake on our part. The text has been revised as follows:

Zoonoses are caused by pathogens naturally transmissible between humans and wild or domestic animals. Places where humans and animals or animal products interact create interfaces that facilitate zoonotic agent transmission. Notably, approximately 99% of endemic zoonotic infections in humans originate from domesticated animals, within anthropogenic environments, either directly or indirectly through contaminated food or vectors (Kock et al. 2022). Morand, et al. provided statistical evidence supporting the positive relationship between the duration of domestication and the diversity of zoonotic agents that humans share with each domestic species, which was initially hypothesised by McNeill. Additionally, over 60% of human emerging infectious diseases (EIDs) are zoonotic (Jones et al. 2008). Although direct zoonotic spillover from wildlife is rare and wildlife-to-human transmission typically occurs through indirect transmission (Kock et al. 2022), more than 70% of these zoonotic emergences are caused by pathogens with a wildlife origin (Jones et al. 2008).

1.7. 62-63: I would add “human behavior” and not only “proximity to the animal species”.

The sentence has been corrected.

1.8. 68-69 “monitor and mitigate potential spillover event” I think we can monitor indicators that can lead to a spillover and effective circulation in different species, more than monitoring the spillover.

We thank Reviewer #1 for this insightful observation. Revised version:

Bridging this gap is crucial for developing effective, locally-relevant strategies (Jones et al. 2013) to monitor and mitigate potential changes in spillover risk that could impact human and animal health.

1.9. 74-75 Without comparisons with other countries, it is not obvious with these figure that Austria is especially a hot spot of biodiversity or a place where interactions between human or wildlife is specifically high.

In this sentence, we present data on biodiversity in Austria, along with statistics on livestock sectors and hunting activities. We would like to clarify that this information does not necessarily indicate Austria as a biodiversity hotspot. Rather, it underscores the existence of the animal-human interface, which may be expanding and requires monitoring at the national level, which is highlighted in this sentence: *“These numbers underline the importance of the human-animal interfaces at the national scale.”* Therefore, we have not made any changes to the sentence. If further changes or clarifications are needed, please let us know.

1.10. 76 Austria adheres to a combination of European and national regulations: which ones? One Health regulations or specific regulation on human health and animal health. What about regulations from WHO and WOA?H?

For example, as a member of the European Union (EU), Austria complies with Directive 2003/99/EC, which mandates the monitoring of zoonoses, zoonotic agents, and related antimicrobial resistance¹. The Directive 2003/99/EC¹ in Austria is translated into the Federal Law on the Monitoring of Zoonoses and Zoonotic Pathogens (Zoonosis Act, “Zoonosengesetz”) (BGBl. I Nr. 128/2005)². Under the framework of Directive 2003/99/EC¹, Regulation (EU) 2017/625³, Commission Implementing Regulation (EU) 2019/627⁴, and Commission Delegated Regulation (EU) 2018/772⁵, Austria reports data on zoonoses and zoonotic agents in animals, food and feed to the European Food Safety Authority (EFSA), which is then integrated into the annual EU One Health Zoonotic Report authored jointly by the EFSA and the European Centre for Disease Prevention and Control (ECDC). Furthermore, the Commission Implementing Decision (EU) 2018/945⁶ provides a list of ~50 communicable diseases and related special health issues to be covered by the epidemiological surveillance network in each country, and for which epidemiological data must be submitted to ECDC annually through the European Surveillance System (TESSy). The Austrian Agency for Health and Food Safety (AGES) is mandated to compile these findings and the aggregated data is subsequently made publicly available. Additionally, Austria has established active monitoring and surveillance programmes primarily targeting *Salmonella*, both in foodstuffs of animal origin and poultry flocks, as required by EU Regulation 2160/2003⁷ and EU Regulation 2073/2005⁸. Furthermore, Regulation (EU) 2022/2371 on serious cross-border threats to health, addresses communicable diseases in humans, including zoonoses, guaranteeing a framework for coordinated epidemiological surveillance, monitoring, and responses⁹.

Moreover, Austria operates within the legal frameworks of the Regulation (EU) 2016/429 on transmissible animal diseases (“Animal Health Law”)¹⁰, which, at national level, refers to the Animal Diseases Act (“Tierseuchengesetz”)¹¹, the Animal Health Act (“Tiergesundheitsgesetz”)¹², and the Bee Disease Act (“Bienenseuchengesetz”)¹³, which govern the surveillance and control of animal diseases, including zoonoses. Additionally, at national level, the Austrian Animal Epidemic Act (“Epidemiegesetz 1950”, BGBl. Nr. 186/1950)¹⁴ regulates notifiable diseases in humans.

Austria also complies with the International Health Regulation, 2005¹⁵, and adheres to the World Organisation for Animal Health (WOAH) International Standards, which consist of The Terrestrial Animal Health Code, The Aquatic Animal Health Code, The Manual of Diagnostic Tests and vaccines for Terrestrial Animals, and The Manual of Diagnostic Tests for Aquatic Animals (see <https://www.woah.org/en/what-we-do/standards/>).

The One Health approach or principles are mentioned in eight EU Regulations, one EU Directive (no longer in force), seven Decisions, and 22 Communications¹⁶. To the best of our knowledge, the term One Health, or its German equivalent “Eine Gesundheit” is not mentioned in the two national laws cited above.

Supporting references:

1 European Commission. Directive 2003/99/EC of the European Parliament and of the Council of 17 November 2003 on the monitoring of zoonoses and zoonotic agents. Official Journal L 325, 12/12/2003 P. 0031-0040, 2003. <https://eur-lex.europa.eu/eli/dir/2003/99/oj>.

2 Bundeskanzleramt Österreich. Bundesgesetz zur Überwachung von Zoonosen und Zoonoseerregern (Zoonosengesetz). 2005. <https://www.ris.bka.gv.at/eli/bgbl/i/2005/128>.

3 European Commission. Regulation (EU) 2017/625 of the European Parliament and of the Council of 15 March 2017 on official controls and other official activities performed to ensure the application of food and feed law, rules on animal health and welfare, plant health and plant protection products, amending Regulations (EC) No 999/2001, (EC) No 396/2005, (EC) No 1069/2009, (EC) No 1107/2009, (EU) No 1151/2012, (EU) No 652/2014, (EU) 2016/429 and (EU) 2016/2031 of the European

Parliament and of the Council, Council Regulations (EC) No 1/2005 and (EC) No 1099/2009 and Council Directives 98/58/EC, 1999/74/EC, 2007/43/EC, 2008/119/EC and 2008/120/EC, and repealing Regulations (EC) No 854/2004 and (EC) No 882/2004 of the European Parliament and of the Council, Council Directives 89/608/EEC, 89/662/EEC, 90/425/EEC, 91/496/EEC, 96/23/EC, 96/93/EC and 97/78/EC and Council Decision 92/438/EEC (Official Controls Regulation). Official Journal L 95, 7.4., 2017. <http://data.europa.eu/eli/reg/2017/625/oj>.

4 European Commission. Commission Implementing Regulation (EU) 2019/627 of 15 March 2019 laying down uniform practical arrangements for the performance of official controls on products of animal origin intended for human consumption in accordance with Regulation (EU) 2017/625 of the European Parliament and of the Council and amending Commission Regulation (EC) No 2074/2005 as regards official controls Official Journal L 131, 17.5, 2019. http://data.europa.eu/eli/reg_impl/2019/627/oj.

5 European Commission. Commission Delegated Regulation (EU) 2018/772 of 21 November 2017 supplementing Regulation (EU) No 576/2013 of the European Parliament and of the Council with regard to preventive health measures for the control of *Echinococcus multilocularis* infection in dogs, and repealing Delegated Regulation (EU) No 1152/2011. Official Journal L 130, 28.5., 2018. http://data.europa.eu/eli/reg_del/2018/772/oj.

6 European Commission. Commission Implementing Decision (EU) 2018/945 of 22 June 2018 on the communicable diseases and related special health issues to be covered by epidemiological surveillance as well as relevant case definitions. Official Journal L 170, 6.7., 2018. http://data.europa.eu/eli/dec_impl/2018/945/oj.

7 European Commission. Regulation (EC) No 2160/2003 of the European Parliament and of the Council of 17 November 2003 on the control of salmonella and other specified food-borne zoonotic agents. Official Journal L 325, 12/12/2003 P. 0001-0015, 2003. <https://eur-lex.europa.eu/eli/reg/2003/2160/oj>.

8 European Commission. Commission Regulation (EC) No 2073/2005 of 15 November 2005 on microbiological criteria for foodstuffs. Official Journal L. 338 22.12.2005, 2005. <http://data.europa.eu/eli/reg/2005/2073/2020-03-08>.

9 European Commission. Regulation (EU) 2022/2371 of the European Parliament and of the Council of 23 November 2022 on serious cross-border threats to health and repealing Decision No 1082/2013/EU. OJ L 314, 6.12.2022, p. 26–63, 2022. <http://data.europa.eu/eli/reg/2022/2371/oj>.

10 European Union. Regulation (EU) 2016/429 of the European Parliament and of the Council of 9 March 2016 on transmissible animal diseases and amending and repealing certain acts in the area of animal health (Animal Health Law). 2016. <http://data.europa.eu/eli/reg/2016/429/2021-04-21>.

11 Bundeskanzleramt Österreich. Tierseuchengesetz (RGBl. Nr. 177/1909). 1909. <https://www.ris.bka.gv.at/GeltendeFassung.wxe?Abfrage=Bundesnormen&Gesetzesnummer=10010172>.

12 Bundeskanzleramt Österreich. Tiergesundheitsgesetz (BGBl. I Nr. 133/1999). 1999. https://www.ris.bka.gv.at/Dokumente/BgblPdf/1999_133_1/1999_133_1.pdf.

13 Bundeskanzleramt Österreich. Bienenseuchengesetz (BGBl. Nr. 290/1988). 1988. https://www.ris.bka.gv.at/Dokumente/BgblPdf/1988_290_0/1988_290_0.pdf.

14 Bundeskanzleramt Österreich. Epidemiegesetz (BGBl. Nr. 186/1950). 1950. https://www.ris.bka.gv.at/Dokumente/BgblPdf/1950_186_0/1950_186_0.pdf.

15 World Health Organization. International Health Regulations (2005). Geneva, 2016. <https://www.who.int/publications/i/item/9789241580410>.

16 Coli, F. & Schebesta, H. One Health in the EU: the next future? *European Papers* 8, 301-316 (2023).

We feel that providing a selected list of international, European, and national regulations would appear incomplete, and delving into the entire regulatory framework under which Austria operates falls outside the scope of our paper. Therefore, after careful consideration, we have decided not to include it in our paper.

1.11. 80 why are the collected data “newly”

We wanted to emphasize that the dataset was novel, i.e., the data was collected specifically for this study and was not available previously. However, we agree with Reviewer #1 that this term is unnecessary, so we have modified the sentence:

In this study, we extracted data from scientific papers and national laboratory reports, spanning 47 years of publications, to generate a real-world network describing the full web of zoonotic interactions in Austria and characterise the various interfaces through which zoonotic spillover may occur.

Results

- Dataset

1.12.94 “were not eligible” need to be a bit more specific (even if it is described in the mat met section).

We thank Reviewer #1 for highlighting this point that may not be clear enough to the reader. Breakdown is detailed in *Supplementary Figure 1: PRISMA flow diagram of publication retrieval and selection*. However, the figure was cited at the end of the paragraph. To enhance clarity, we have modified the sentence as follows:

The search identified 2,186 publications. After 542 duplicates were removed, 1,644 publications were screened with 1,269 excluded at the title/abstract screening stage as they were not eligible (see Supplementary Fig. 1 for a breakdown based on exclusion criteria).

1.13.102-108 the increase level of publications on zoonosis in Austria or worldwide is not informative itself if you do not compare to the general increase of publications between the two period (for example publications with health as a key word increased by x7 between the 2 periods)

As suggested by Reviewer #1, we have conducted a PubMed search spanning the study period using the term "health." The result showed a 6.8-fold increase in the overall number of publications globally, indicating a disproportionate increase in zoonotic disease research compared to general health research over the study period. The corresponding dataset (downloaded from PubMed) was added to the Supplementary Information 1 and to the figshare repository, and the R script was modified accordingly, ensuring reproducibility of this result.

New sentence:

However, a PubMed search using the terms "health" on the same period, yielded 5,791,763 results indicating a mere 6.8-fold increase in health publications globally. This result suggests a disproportionate rise in zoonotic disease research, both at the national and international levels, compared to general health studies.

- Research trends

1.14.123-125 It would be interesting to separate vectors from environment and food to describe the trends to see if the most gradual increase is due to publication in environmental matrix or in vectors.

We agree with Reviewer #1 and have added a new figure in the file Supplementary Information 2:

Supplementary Figure 4: Trends in research interest on zoonotic agents in Austria within the environmental compartment, 1975-2022, as measured by the number of investigations. Solid lines show fitted trend (loess regression); shaded areas represent the corresponding 95% confidence interval.

This new figure disaggregates data within the environmental compartment, offering a detailed breakdown of the categories and enabling a more specific analysis of the observed trend in the number of investigations. Notably, it shows that the observed increase is related with an increase in the investigations of vectors. These results and the supplementary figure are mentioned in the main text:

The environmental aspect (including environmental media, plant-based food, and vector) was not considered in studies on zoonotic diseases in Austria until 1997 but subsequently demonstrated the most gradual increase in scientific interest (Fig. 1b), primarily driven by a rise in investigations on vectors (Supplementary Fig. 4).

- Zoonotic web actors

1.15. 132-134 an additional comparative analysis of zoonotic richness between human, non-human mammals and birds would be useful.

This is a good point. We have updated Figure 2 accordingly. The revised version now includes panel b, which illustrates the zoonotic agent richness per taxonomic class, with a breakdown between non-human mammals and human. Panel c (previously panel b) displays the zoonotic agent richness per order.

The caption of Figure 2 has been revised accordingly:

Figure 2: a. Bubble plot illustrating the distribution of the zoonotic agent genera across Austrian hosts, grouped by genus, 1975-2022. Only publications that investigated naturally occurring zoonotic infections were considered. Node size corresponds to the number of zoonotic agents detected within a specific genus during this timeframe. Colours correspond to the zoonotic agent superkingdom. **b. Zoonotic agent richness per host taxonomic class**, with data disaggregated for human and non-human mammals. **c. Zoonotic agent richness per host taxonomic order.** When host scientific or common name was not specific enough within the publication, the taxonomic order cannot be retrieved and the host name, as mentioned in the publication, was used (e.g., lizard, new-world camelid).

The R script has also been modified to ensure reproducibility of the results and figures.

- Imported and emerging zoonotic agents

1.16. 243 “emerged in the past 20 years”, I would say “where found to emerge” since this is related to existing publications and depends of the dataset.

This sentence has been corrected. Revised version:

Notably, all of them were found to emerge in the past 20 years.

Discussion

1.17. 254 : I would add “including human behavior and demographic and global changes”

This is a good point, thank you. We have modified the sentence as follows:

Cross-species transmission and emergence of zoonotic-origin diseases occur at complex animal-human-environment interfaces, within dynamic social-ecological systems influenced by human behaviour, demographic shifts, and global changes.

1.18.257 why “novel” dataset for a bibliographical dataset

The adjective has been removed. Revised sentence:

Here, we present the first attempt to analyse nearly 50 years of data on naturally occurring zoonotic infections (or contaminations) in Austria, leveraging an original One Health approach based on network theory.

1.19. 280 coronavirus are not in the dataset of emerging zoonotic agents, it is discussed later in the paper but it emphasize the fact that the dataset selection using “zoono” as a criteria has some limits since papers can described emergence without mentioning zoonosis in the text. Other

emergences could have been missed, therefore it is difficult to conclude on “most emerging pathogens in Austria are Bacteria and helminths”.

We fully agree with this comment. In the revised version, we mention that to expand further the dataset, a more specific search should be performed:

Future extensions could also explore conducting targeted searches for each zoonotic agent, based on available global lists (Olival, K. J. et al, 2017; Carlson et al. 2022; Wardeh et al., 2015), in the literature and international health organisation websites (e.g., World Health Organization, World Organisation for Animal Health).

Additionally, in the discussion, we have moderated our statement to reflect this perspective. Revised sentence:

While there is often an emphasis on viral emergence, particularly considering that RNA viruses pose the most significant threat, our findings offer a different perspective. Within our dataset, six out of eight emerging pathogens in Austria were bacteria and helminths.

1.20. 293 many sources (sero) positive, It would be nice to have in the results and discussion (and not only in the methodology section) some explanation and analysis about the papers found to described either seropositive animals/humans or with pathogen identification. It has some impact on robustness of the results regarding types of emergence and on surveillance strategies.

We thank Reviewer #1 for this valuable comment and agree we should provide more details on this aspect. We have therefore added a descriptive analysis of the methods used for zoonotic agent detection. The new Table 1 provides a breakdown of the investigation types and their numbers categorised by host taxonomic class. The R script for the analysis has also been updated to reflect these changes.

New text (Results → Zoonotic web actors and interfaces):

In 78.6% (777 out of 989) of the host investigations, direct detection of zoonotic agents was achieved and represented the preferred method for bacteria and eukaryotes across all investigated host taxonomic classes. Conversely, viral circulation was primarily evidenced by indirect methods detecting antibodies (Table 1).

Caption of the new Table 1:

Table 1: Breakdown of the methods used for zoonotic agent detection and number of investigations by host taxonomic class and zoonotic agent superkingdom, Austria, 1975-2022. The class Mammalia is further disaggregated for humans and non-human mammals.

Furthermore, we have added a new sentence in the first paragraph of the Discussion:

With approximately 80% of animal investigations and all investigations on vectors, food, and environmental matrices supported by direct evidence of zoonotic agents, we are confident in the robustness of our results.

1.21. 296 replace “demonstrated” by “observed”.

The verb “demonstrate” has been replaced by “observe”.

1.22.306 Can you really conclude on the driving forces of anthropogenic activities with the grouping of most food products into one community. Do you have enough data on food product to conclude?

This is a great question, thank you. In our analysis, food-related nodes comprise 15.8% (31 out of 196) of the total nodes in the unipartite network of zoonotic agent sharing. In our study, we used the Leiden algorithm (Traag et al., 2019), which relies on a measure called modularity (Newman & Girvan, 2004).

The method aims to optimise modularity by maximising the difference between the actual and expected number of edges within communities. Therefore, the model does not “know” the type of the nodes, only edge attributes are utilised, and the proportion of nodes identified as “food” does not influence community patterns. Although node attribute can be used to enhance community detection on networks (Bianconi et al., 2009; Jia et al., 2017), in our study, we did not implement this approach. To enhance clarity, this aspect has been better specified in the Methods section (→ Community detection):

We used the Leiden algorithm (Traag et al. 2019), which relies on a measure called modularity (Newman & Girvan, 2004), to detect communities of zoonotic agent sharing within the research-adjusted one-mode network of zoonotic sources. The method aims to optimise modularity by maximising the difference between the actual and expected number of edges within communities. The Leiden algorithm is considered as an improvement over the Louvain algorithm (Blondel et al., 2008).

Concerning our interpretation of the community pattern and the role of anthropogenic factors, our analysis, visualized in Fig. 5 and Supplementary Fig. 8 (previously Supplementary Fig. 7), shows that zoonotic agents shared between sources in this community are primarily foodborne pathogens. Moreover, the literature reports the importance of the processing and transformation step in food contamination. To make our statement stronger, we have added some supporting references (see end of this answer). Revised sentence in the Discussion:

The grouping of most food products into one community, predominantly sharing zoonotic agents typically associated with foodborne infections (EFSA & ECDC, 2023; Hald et al., 2007) (21/24, 87.5%, including the five leading causes of foodborne diseases in the EU: Campylobacter, Salmonella, Yersinia, E. coli and Listeria (EFSA & ECDC, 2023), as well as 10 serovars of Salmonella enterica subsp. enterica), suggests that anthropogenic activities, particularly those related to food processing and transformation (Sun et al., 2021; Gallo et al., 2020), may further influence the pattern of assembly within zoonotic source communities.

Regarding significance of the communities: The aim of community detection on networks is to identify the clusters and, possibly, their hierarchical organisation, by only using the information on the graph topology (Fortunato, 2010). Despite the development of numerous community detection algorithms (Fortunato, 2010; Orman, et al., 2012; Jebabli, et al. 2018), the challenge of distinguishing real communities from “pseudo-communities” remains unresolved (He, et al., 2021). Many algorithms are not able to distinguish them and find clusters in random graphs as well, despite the lack of significance. The assessment of cluster significance is a burgeoning area of research (Lancichinetti, et al. 2011; Bianconi, et al. 2009). Some works define significance based on the robustness or stability of a partition against perturbations in the graph structure. In other words, a significant partition persists even when the graph undergoes slight modifications. Conversely, insignificant partitions are easily disrupted by even minor changes, leading to different clustering. While this area of research offers valuable insights, it falls outside the scope of our current paper, and we did not implement this approach nor discuss this in the discussion.

Supporting references:

- Bianconi, G., Pin, P. & Marsili, M. Assessing the relevance of node features for network structure. PNAS 106, 11433-11438 (2009).
- Blondel, V. D., Guillaume, J.-L., Lambiotte, R. & Lefebvre, E. Fast unfolding of communities in large networks. J. Stat. Mech. Theory Exp. 2008, P10008 (2008).
- Fortunato, S. Community detection in graphs. Physics Reports 486, 75-174 (2010).
- He, Z., Chen, W., Wei, X. & Liu, Y. On the statistical significance of communities from weighted graphs. Sci. Rep. 11, 20304 (2021).
- Jebabli, M., Cherifi, H., Cherifi, C. & Hamouda, A. Community detection algorithm evaluation with ground-truth data. Physica A: Statistical Mechanics and its Applications 492, 651-706 (2018).
- Jia, C., Li, Y., Carson, M. B., Wang, X. & Yu, J. Node Attribute-enhanced community detection in complex networks. Sci. Rep. 7, 2626 (2017).
- Lancichinetti, A., Radicchi, F., Ramasco, J. J. & Fortunato, S. Finding statistically significant communities in networks. PLoS One 6, e18961 (2011).
- Orman, G. K., Labatut, V. & Cherifi, H. Comparative evaluation of community detection algorithms: a topological approach. J. Stat. Mech. Theory Exp. 2012, P08001 (2012).
- Traag, V. A., Waltman, L. & van Eck, N. J. From Louvain to Leiden: guaranteeing well-connected communities. Sci. Rep. 9, 5233 (2019).
- Newman, M. E. J. & Girvan, M. Finding and evaluating community structure in networks. Phys. Rev. E 69, 026113 (2004).

European Food Safety Authority & European Centre for Disease Prevention. The European Union One Health 2022 Zoonoses Report. EFSA Journal 21, p211202 (2023).

Hald, T. et al. World Health Organization estimates of the relative contributions of food to the burden of disease due to selected foodborne hazards: a structured expert elicitation. PLoS One 11, e0145839 (2016).

Sun, L. et al. Epidemiology of foodborne disease outbreaks caused by nontyphoidal Salmonella in Zhejiang Province, China, 2010-2019. Foodborne Pathog Dis 18, 880-886 (2021).

Gallo, M., Ferrara, L., Calogero, A., Montesano, D. & Naviglio, D. Relationships between food and diseases: What to know to ensure food safety. Food Research International 137, 109414 (2020).

2. Reviewer #2

(Remarks to the Author):

The authors of this study investigated the network of hosts sharing microbiological agents identified as zoonotic in literature published in Austria from 1975 to 2022. While the study reveals the complexity of a microbiological agent-sharing network amongst various hosts based on the vast datasets, the manuscript requires revision with particular attention to the following points.

We thank Reviewer #2 for their overall positive comments. We have addressed each of them below.

2.1. First, in the zoonotic web the authors constructed, edges represent the sharing of microbiological agents between hosts. For these agents and hosts, "the most specific NCBI-resolved zoonotic agent and host names" are selected and matched during network construction. However, the articles that formed the basis of this network construction were selected only when they included "Zoono*". I wonder if this process would have removed articles whose information could form edges according to the process of network construction but were removed because they simply did not define the agents studied as zoonotic. Would the network constructed based on literature search using a list of zoonotic microbiological agents, instead of using "Zoono*", be different from the network presented in the manuscript?

In our study, we collected data on source [host-food-vector-environment]-agent interactions from publications that acknowledged these interactions as zoonotic. The scientific and common names of each host, vector, and agent were validated against the NCBI Taxonomy database, as part of the cleaning/validation process. Then, the network was built using NCBI-validated names. All interactions that were collected through the review process were kept (no edge was removed). Assessing, without the data available, whether a network constructed solely based on literature search using a list of zoonotic microbiological agents, as opposed to employing the "Zoono*" search term, would yield different results is challenging. However, we believe our chosen approach offers added value by highlighting gaps in zoonotic research, such as SARS-CoV-2 not being identified as zoonotic in any study conducted in Austria.

Another, or complementary, strategy for collecting literature data could be to use a global list of known zoonotic agents (e.g., Mollentze & Streicker, 2020; Olival et al., 2017, Carlson et al., 2022; Wardeh et al., 2015) and search specifically for each of them in the Austrian literature. However, this search may require significant time investment and resources, and utilising text mining tools for web search, as suggested by Reviewer #1, could enhance efficiency. These points have been summarised in the revised version of the Discussion (→ last paragraph describing the study limitations):

Ultimately, broadening the dataset by including additional data on natural infections documented in diverse laboratories (e.g., university laboratories that often investigate a broader range of sources and agents compared to national reference labs) as well as event-based surveillance (EBS) data sourced from various, non-official channels (Dub et al., 2023), could significantly enrich the dataset and enhance the depth of the analysis. (...) Future extensions could also explore conducting targeted searches for each zoonotic agent, based on available global lists (Olival, K. J. et al, 2017; Carlson et al. 2022; Wardeh et al., 2015), in the literature and international health organisation websites (e.g., World Health Organization, World Organisation for Animal Health). However, the latter may necessitate considerable time and resources; employing automated data extraction methods and tools could improve efficiency (Gutiérrez-Sacristán, A. et al., 2017; Paraskevopoulos et al., 2022; Walker et al., 2022).

Supporting references:

Carlson, C. J. et al. The Global Virome in One Network (VIRION): an atlas of vertebrate-virus associations. *mBio* 13, e02985-02921 (2022).

Dub, T. et al. Epidemic intelligence activities among national public and animal health agencies: a European cross-sectional study. *BMC Public Health* 23, 1488 (2023).

Mollentze, N. & Streicker, D. G. Viral zoonotic risk is homogenous among taxonomic orders of mammalian and avian reservoir hosts. *Proc Natl Acad Sci U S A* 117, 9423-9430 (2020).

Olival, K. J. et al. Host and viral traits predict zoonotic spillover from mammals. *Nature* 546, 646-650 (2017)

Wardeh, M., Risley, C., McIntyre, M. K., Setzkorn, C. & Baylis, M. Database of host-pathogen and related species interactions, and their global distribution. *Sci. Data* 2, 150049 (2015).

2.2. Second, the authors attempted to reduce bias from research efforts by accounting for the number of studies for the same sources and agents. While this might address the quantity of investigations, would this also address the presence or absence of investigations on particular sources or agents? Almost every network property seems centered around humans, followed by livestock and companion animals. Does the approach used in this study address research efforts not conducted ever due to a lack of interest? (although I also acknowledge that this could indicate a potential absence of zoonotic agents and thus no edge). The manuscript requires an in-depth discussion of potential limitations in their network construction processes.

We thank Reviewer #2 for raising this interesting point. In this work, we did not address research efforts not conducted ever due to a lack of interest because the true size of the unsearched and undiscovered is unknown. This limitation is now underscored in the Discussion section, highlighting the inherent constraints of our analysis:

Furthermore, despite efforts to control for research bias, our analysis is inevitably constrained by the existence of zoonotic source-agent associations that are either unknown or not yet published. This constitutes a major challenge in our understanding of zoonotic interactions.

Regarding network datasets, incompleteness is often inevitable due to measurement limitations or the dynamic nature of networks (i.e., the dataset changes over time). Additionally, acquiring comprehensive network data can be costly or unfeasible. Consequently, missing topology structure, which means that edges and nodes are absent, is common in networks. These missing data can be categorised into four types, each posing unique challenges for prediction tasks:

- Predicting missing links
- Predicting missing link attributes
- Predicting missing node attributes
- Predicting missing node.

However, a fundamental challenge in link prediction for biological data lies in interpreting the association between entities. E.g., if a host and a zoonotic agent are predicted as associated, what implications does this predicted association hold from both biological and epidemiological perspectives? In response to Reviewer #2 comment, we have elaborated on the network completion problem in the Discussion section to suggest avenues for future research.

New section in the discussion:

Alternative methodologies have been employed to investigate spillover events. For instance, Grange, et al. ranked the spillover risk from known and newly discovered wildlife-origin viruses using a database of wildlife host-virus association combined with expert opinion on drivers of spillover. Washburne, et al. utilised percolation models to analyse cross-species transmission, uncovering inherent nonlinearity in spillover rate. Additionally, Olival, et al. used a dataset of mammal host-virus associations as a proxy for measuring spillover and generalised additive models (GAMs) to identify predictors of host viral richness and estimate the number of undiscovered viruses for each mammal species. Missing or unobserved links and nodes frequently occur in collected network data (Lü et al., 2011), which can impact the network properties. Diverse methods have been proposed to infer missing links (Kim & Leskovec, 2011; Clauset et al., 2008; Guimerà & Sales-Pardo, 2009) and nodes (Kim & Leskovec, 2011; Liu & Zhou, 2011). Notably, edge prediction accuracy can be enhanced through the use of network community structure (Yan & Gregory, 2012). These methods offer valuable mathematical or statistical approaches that could be applied in future research leveraging our dataset and analytical framework to assess the spillover risk and infer the presence of a zoonotic agent in a source where data has been lacking.

Supporting references:

Clauset, A., Moore, C. & Newman, M. E. J. Hierarchical structure and the prediction of missing links in networks. *Nature* 453, 98-101 (2008).

Grange, Z. L. et al. Ranking the risk of animal-to-human spillover for newly discovered viruses. *PNAS* 118, e2002324118 (2021).

Guimerà, R. & Sales-Pardo, M. Missing and spurious interactions and the reconstruction of complex networks. PNAS 106, 22073-22078 (2009).

Kim, M. & Leskovec, J. in Proceedings of the 2011 SIAM International Conference on Data Mining (SDM). 47-58.

Lü, L. & Zhou, T. Link prediction in complex networks: A survey. Physica A: Statistical Mechanics and its Applications 390, 1150-1170 (2011).

Olival, K. J. et al. Host and viral traits predict zoonotic spillover from mammals. Nature 546, 646-650 (2017).

Washburne, A. D. et al. Percolation models of pathogen spillover. Philosophical Transactions of the Royal Society B: Biological Sciences 374, 20180331 (2019).

Yan, B. & Gregory, S. Finding missing edges in networks based on their community structure. Phys. Rev. E 85, 056112 (2012).

2.3. There needs to be clearer explanations or discussion on the epidemiological meaning of individual network features. For example, some of the indirect connections in the giant connected network component might not have epidemiological implications if the transfer of zoonotic agents is not possible due to immunological barriers.

We thank Reviewer #2 for this insightful comment. We have tried to thoroughly address it throughout the manuscript.

First, clear explanations of the biological meaning of the links in the zoonotic web have been incorporated into the Methods section (→ Analysis of the zoonotic web):

A link between a zoonotic agent i and a vertebrate host j indicates that agent i was directly or indirectly detected in host j . A link between a zoonotic agent i and a vector j signifies that agent i was identified in vector j , implying that vector j may transmit agent i to a vertebrate host through a bite or mechanically. Likewise, a link between a zoonotic agent i and an environmental matrix j indicates the presence of agent i in environment j , potentially leading to infection of a vertebrate host upon contact. Lastly, a link between a zoonotic agent i and a food matrix j indicates that agent i was detected in food j , which may result in infection of vertebrate hosts through ingestion.

Additionally, to enhance clarity and allows Figure 1 to be self-explanatory, we have included these explanations in the caption of Figure 1, along with a reference to the colour coding used in the graph:

Figure 3: Network representation of the zoonotic web in Austria, 1975-2022. *This representation uses the D3 forceLink layout, providing a comprehensive visualisation and offering valuable epidemiological insights into naturally occurring zoonotic interactions in Austria. The zoonotic web is a bipartite network, where each node (circle) represents an actor in the zoonotic web in Austria, 1975-2022, with one set of nodes representing zoonotic agents and the second set representing zoonotic sources that belong to different categories: hosts, vectors, foodstuffs, and environmental matrices. A link between a zoonotic agent i (black nodes) and a vertebrate host j (red nodes) indicates that agent i was directly or indirectly detected in host j ; a link between a zoonotic agent i and a vector j (yellow nodes) signifies that agent i was identified in vector j , implying that vector j may transmit agent i to a vertebrate host through a bite or mechanically; a link between a zoonotic agent i and an environmental matrix j (green nodes) indicates the presence of agent i in environment j , potentially leading to infection of a vertebrate host upon contact; and a link between a zoonotic agent i and a food matrix j (blue nodes) indicates that agent i was detected in food j , which may result in infection of vertebrate hosts through ingestion. Node size represents the actor's degree and is coloured by the category the actor belongs to. The bottom-right graph illustrates the degree distribution for the “zoonotic agents” and “zoonotic sources” partitions, the latter being disaggregated based on source categories. The node degree centrality for each zoonotic source corresponds to the zoonotic agent richness, i.e., the number of taxa directly or indirectly evidenced from the zoonotic source. The node degree centrality for each zoonotic agent corresponds to the zoonotic source range, i.e., the number of sources from which the agent has been directly or indirectly evidenced, reflecting its “host” or “zoonotic source” plasticity.*

Within the zoonotic web, all links between the zoonotic sources and the agents represent realised interactions, i.e., edges included in this graph (Figure 3) have been described in the publications used to compile the dataset and are therefore of epidemiological relevance. For this reason, we would respectfully disagree with Reviewer #2 regarding potential immunological barriers. However, it is important to highlight that by transforming the zoonotic source-agent bipartite network into a source-

source unipartite network, a "transmission-potential network" is created, where hosts are linked based on shared zoonotic agents (Pilosof et al., 2015). We have therefore added clarification in the Methods section (→ Analysis of the zoonotic web) to specify that these "indirect interactions" represent "potential" transmissions:

The zoonotic source-agent network was subsequently projected into a one-mode network of zoonotic agent sharing among sources. Edges were weighted by the number of shared zoonotic agents between two sources. By transforming the zoonotic source-agent bipartite network into a source-source unipartite network, a "transmission-potential network" is created, where hosts are linked based on shared zoonotic agents (Pilosof et al., 2015).

We have additionally revised the mention of the transmission chain in the Results and Discussion sections and have added the adjective "potential" when relevant:

Results (→ network of zoonotic agent sharing), revised sentences: *Thus, for one zoonotic agent, connected sources belong to the same potential transmission chain (VanderWaal et al., 2014; Pilosof et al., 2015) (Fig. 4a).*

Discussion, revised sentence: *We argue that the comprehensive analysis of the zoonotic web holds greater value when studying potential zoonotic transmission chains compared to the commonly employed host-pathogen network approach, as it offers a broader epidemiological perspective and more analytical flexibility.*

Following the ecological network model proposed by Simmons et al. (2019), indirect interactions illustrate the influence of one zoonotic source on another, mediated by the zoonotic agent. For instance, if a zoonotic agent is identified in two different sources, its prevalence in one source may impact its prevalence in the other. These indirect interactions are depicted in the unipartite network of zoonotic agent sharing, where connections between zoonotic sources indicate shared zoonotic agents.

Finally, following Reviewer #2 comment, we have added an additional discussion on this specific aspect:

Within the zoonotic web, multiple zoonotic sources contribute to the maintenance and spread of zoonotic agents. Studying the source-source network of zoonotic agent sharing is necessary to reveal indirect interactions (Simmons et al., 2019), where one source influences another through shared agents. For example, if an agent is found in two sources, its prevalence in one may affect the other. However, these indirect interactions may lack epidemiological significance if, for instance, physical barriers prevent agent transfer between sources, such as when the sources do not share similar ecological niches (Plowright et al., 2017).

Supporting references:

Pilosof, S., Morand, S., Krasnov, B. R. & Nunn, C. L. Potential parasite transmission in multi-host networks based on parasite sharing. *PLoS One* 10, e0117909 (2015).

Plowright, R. K. et al. Pathways to zoonotic spillover. *Nat. Rev. Microbiol.* 15, 502-510 (2017).

Simmons, B. I. et al. Motifs in bipartite ecological networks: uncovering indirect interactions. *Oikos* 128, 154-170 (2019).

VanderWaal, K. L., Atwill, E. R., Isbell, L. A. & McCowan, B. Quantifying microbe transmission networks for wild and domestic ungulates in Kenya. *Biol. Conserv.* 169, 136-146 (2014).

The following are specific comments:

2.4. Line 81: The full web can be misleading because the network construction is based on literature review, which has some limitations as mentioned above.

Thank you for noticing this point. We agree the use of "full" can be misleading and the term has been removed. In the Supplementary Figures 10-13, which show subgraph of the "full" network, the term "full" was replaced by "original".

2.5. Line 93: Although explained in the method section, it would be good to mention briefly the screening terms used for literature review here.

We appreciate Reviewer #2's suggestion to reiterate the search terms in the Results section for clarity, yet, we must consider the word limitations outlined in the authors' guidelines. Our search strategy involved two approaches: one across four scientific paper databases, using the query ("Zoono*" AND ("Austria" OR "Österreich")), and the other within the database of the Austrian Agency for Health and Food Safety (AGES), using the keyword "Zoono*" exclusively. Given that our paper has already been significantly expanded during the revision process, we would prefer to avoid duplicating information already mentioned in the manuscript.

2.6. Line 136: If USUV is not mentioned earlier, please write the full name here.

We thank Reviewer #2 for identifying this oversight. We have corrected the omission by providing the full name of Usutu virus (USUV) in this instance.

2.7. Line 149: I would use Figure S6 in the main text (with/without Figure 3). Figure S6 is more intuitive. In Figure S6, please highlight edges between major sources and agents (e.g., edges between the top 10 sources and agents) in bold lines.

We appreciate Reviewer #2's comment and suggestions. After thorough consideration, we have decided to maintain Figure 3 as it is, while improving Supplementary Fig. 7 (formerly Supplementary Fig. 6) as recommended. Our rationale includes acknowledging the "spatial" grouping of zoonotic sources and agents in Figure 3, which enhances the representation of potential transmission chains. Additionally, Figure 3 provides node degree information, which aligns with our preferred metrics for the unweighted bipartite network.

To further enhance clarity, we have developed an interactive user-friendly dashboard, which we believe significantly improves visualization and communication of our findings. The link to the dashboard (<https://vis.csh.ac.at/zoonotic-web/dashboard.html>) has been added to the captions of Figure 3 and Figure 5, along with a password for access until the manuscript is published (User: csh; Password: zoonotic-web-vis). We plan to release the dashboard upon publication, after coordinating with the Editorial Board on the timing.

Furthermore, following Reviewer #2's recommendations, Supplementary Figure 7 (formerly Supplementary Fig. 6) was improved by adding a similar colour scheme for the source nodes as in Figure 3. Additionally, the top 10 sources and zoonotic agents based on node degree centrality have been highlighted in darker colours. Finally, links between the top 10 sources and the agents they interact with have been emphasised. Caption for Supplementary Fig. 7 has been adapted accordingly:

Supplementary Figure 7: Conventional representation of the bipartite network of zoonotic interactions ("zoonotic web"), Austria, 1975-2022. The lower set of nodes represents zoonotic agents, while the upper set represents zoonotic sources categorised into hosts (red), vectors (yellow), foodstuffs (blue), and environmental matrices (green). The top 10 sources and zoonotic agents, based on node degree centrality, are highlighted with colours corresponding to their node category. Edges between the top 10 sources and the zoonotic agents they interact with are highlighted using the colour of the respective zoonotic source.

2.8. Lines 148 onwards: Define network metrics briefly, although they are defined in the method, to help readers without knowledge of network analysis.

Definitions for the four centrality metrics have been added to this paragraph and the following one:

The analysis of the zoonotic web showed a right-skewed distribution of the node degree centrality (the number of links a node has), revealing few nodes with a high number of connections whereas most of the nodes had one.

In this network, node rankings using the four centrality metrics (degree; strength, i.e., the sum of the weights of edges to/from a node; betweenness, i.e., the number of shortest paths that go through a node; and closeness, i.e., the average distance to all other nodes) showed positive correlation ($0.26 < \text{Kendall's Tau} < 0.77$, $p < 0.001$ in all cases, Supplementary Table 5).

2.9. Line 181: "Research effort-adjusted" is too vague and might give the impression that there is no bias unadjusted. Briefly explain how this is done when this is first mentioned in the manuscript.

Thank you for this suggestion, the text has been revised accordingly:

We generated a unipartite scientific research effort-adjusted network of zoonotic sources (i.e., accounting for research biases), based on zoonotic agent sharing.

2.10. Line 187: List and briefly define the list of four centrality metrics.

The list and brief definition of the centrality metrics has been added. Node degree centrality has been defined in the previous paragraph (see answer to comment 2.8):

In this network, node rankings using the four centrality metrics (degree; strength, i.e., the sum of the weights of edges to/from a node; betweenness, i.e., the number of shortest paths that go through a node; and closeness, i.e., the average distance to all other nodes) showed positive correlation ($0.26 < \text{Kendall's Tau} < 0.77$, $p < 0.001$ in all cases, Supplementary Table 5).

2.11. Line 200: What is the difference in epidemiological meaning between closeness and degree centrality metrics?

On lines 478-479 of the submitted manuscript (section Methods), we defined degree centrality as follows: (...) *degree centrality (the number of links a node has) (...). Within the network context, the node degree centrality for each zoonotic source corresponds to the zoonotic agent richness, i.e., the number of taxa directly or indirectly evidenced from the zoonotic source. Similarly, the node degree centrality for each zoonotic agent corresponds to the zoonotic source range, i.e., the number of sources from which it has been directly or indirectly evidenced, reflecting its "host" or "zoonotic source" plasticity.*

On lines 200-202 of the submitted manuscript, in the section describing results on the source-source unipartite network, we explained closeness centrality as follows: *Closeness centrality identifies nodes that are "close" to many other nodes; therefore, zoonotic sources which share numerous zoonotic agents with numerous sources would have high closeness centrality.*

To better highlight the epidemiological significance of these two measures, we have rephrased the first statement as follows:

In the epidemiological context, the node degree centrality for each zoonotic source corresponds to the zoonotic agent richness, i.e., the number of taxa directly or indirectly detected in the zoonotic source. Similarly, the node degree centrality for each zoonotic agent corresponds to the zoonotic source range, i.e., the number of sources from which it has been directly or indirectly evidenced, reflecting its "host" or "zoonotic source" plasticity.

Furthermore, we have added the following sentence, with supporting reference, in the Results section (→ Network of zoonotic agent sharing, after the results of the Kendall correlation):

Degree and strength centrality reflect the direct co-sharing pattern of zoonotic agents among sources. In contrast, betweenness and closeness centrality provide insights into indirect sharing through other sources (Gómez et al., 2013)

Supporting reference:

Gómez, J. M., Nunn, C. L. & Verdú, M. Centrality in primate–parasite networks reveals the potential for the transmission of emerging infectious diseases to humans. PNAS 110, 7738-7741 (2013).

Reviewer #3

(Remarks to the Author):

3.1. This paper described the analysis of zoonotic pathogen spillovers between human-animal and environment in Austria, based on published and grey literature. Understanding the spillover mechanisms and links with human practices is key to develop target and risk based surveillance and to prevent zoonotic disease emergence at source. Moreover the work is original by the method used as the authors have used network analysis and One-Health 3 cliques to characterise the spillover events.

This work present important and significant results, highlighting the need to assess the real figures of zoonotic agents circulating in the territory, showing the recent consideration of the environment component in disease transmission research, showing that beyond viral threats, disease emergence caused by bacteria has been overlooked.

The work is also strong as it highlights its own limits, mentioning that the results might be skewed towards human, and domesticated animals as they have been the focus of research, compared to other species and pathogens less studies. Nevertheless the result remains relevant and interesting, the application of such an approach into other setting e.g. hot spot developing countries would be very valuable to highlights similar gaps in understanding but also to demonstrate how our epidemiological knowledge which is used for health strategy decision making is highly skewed based on research data available (and therefore funding priorities) which do not always reflect the field reality. A mention to this potential application in the conclusion and perspective would further improve this publication.

We thank Reviewer #3 for their positive feedback on our manuscript. We are glad they liked our work.

Moreover, we are grateful to Reviewer #3 for their suggestion regarding the conclusion. The following sentence has been added to the revised manuscript:

Applying this approach across different settings, particularly in regions identified as hotspots for zoonotic disease emergence, can expose critical knowledge gaps and reveal how existing epidemiological understanding, shaped by research data availability and funding priorities, may not always reflect on-the-ground realities.

Additional comments:

3.2. As the authors highlight very well that the data might be skewed to zoonotic sources closer to humans; it might be worthwhile to mention that the data might also be skewed towards zoonotic agents which have been priorities internationally as pandemic threats and therefore highly studied and under high active surveillance (e.g. Avian Influenza- poultry, Rabies-dogs, BSE-cattle).

We thank Reviewer #3 for this comment: We have strengthened this aspect into the Discussion section, providing concrete examples for diseases included in our dataset:

Ten genera of zoonotic agents constituted 41% of the published research on zoonotic diseases in Austria, with seven of them involving agents subjected to compulsory surveillance and reporting in humans and/or animals (Bundeskanzleramt Österreich, 2005). This outcome underscores an imbalance in research interest, likely influenced by funding opportunities as well as global- and national-level prioritisation, typically based on known incidence and potential impact on human populations. Notably, diseases under European regulatory surveillance, such as those responsible for foodborne outbreaks, or those posing a threat to global public health, like influenza A virus, tend to receive more attention. Such a bias may lead to a skewed assessment of the overall zoonotic risk, especially concerning potentially "neglected" zoonoses such as certain helminth infections (e.g., dirofilariosis, dicrocoeliosis, hepatic capillariosis).

Supporting references:

Bundeskanzleramt Österreich. Bundesgesetz zur Überwachung von Zoonosen und Zoonoseerregern (Zoonosengesetz). 2005. <https://www.ris.bka.gv.at/eli/bgb/I/2005/128>.

3.3. The authors mention page 9 and 10 from line 338 that one crucial element of surveillance to prevent emergence at community level is the use of sentinel - It is important to take into perspective that sentinels, as an active component of surveillance allows to generate additional knowledge and better understanding of pathogens circulation and spillover, might also allow to early detect an infection but only if the event based component of the community surveillance is effective, I.e. the communities are aware of risks and trained on how to recognize them and undertake quick action to prevent spillover or spread and alert relevant stakeholder to take action. This should be specified in the manuscript - this work requires also tailoring to the specific socio-economic context where the pathogen evolves, which ties in nicely with this work. Reference to the recent work of Guenin et al. PLOS NTD could be made.

This is a very valid point, thank you.

Please, note that in the discussion, the term “communities” refers to the network communities, not to community-based surveillance. The term has been better specified in the text:

Selecting sentinels that are distant from each other in the network proved to enhance the overall probability of one sentinel being in proximity to an outbreak, thereby increasing the likelihood of detection. For example, distributing the sentinels in different network communities and prioritising surveillance (...).

We have also adapted this section of the Discussion to better reflect the importance of the socio-economic context, notably the community perception of the disease. The suggested reference was very helpful, we thank Reviewer #3 for the suggestion. The important role sentinels may play in active surveillance and early detection was already addressed in the Discussion (see below in bold); therefore, we did not expand further on this aspect.

Modified sentence:

*Identifying sentinels through network metrics should depend on the topology of the network, the infectious agent to be monitored (e.g., endemic versus emerging, transmission route(s)), the (estimated) infection rate, the target population, the **objective of the surveillance (e.g., early detection versus prevalence estimation)** (LoGiudice et al., 2003; Markotter et al., 2023), and the specific epidemiological, ecological, and socio-cultural-economic context (e.g., what resources are available, what measures are acceptable, what is the community perception of the disease (Guenin et al., 2022)).*

Supporting references.

Guenin, M.-J. et al. A participatory epidemiological and One Health approach to explore the community's capacity to detect emerging zoonoses and surveillance network opportunities in the forest region of Guinea. PLoS Negl. Trop. Dis. 16, e0010462 (2022).

LoGiudice, K., Ostfeld, R. S., Schmidt, K. A. & Keesing, F. The ecology of infectious disease: effects of host diversity and community composition on Lyme disease risk. Proc Natl Acad Sci U S A 100, 567-571 (2003).

Markotter, W. et al. Prevention of zoonotic spillover: From relying on response to reducing the risk at source. PLoS Pathog. 19, e1011504 (2023).

Other changes

Here are the changes made following discussions with colleagues after publication of our preprint:

- To increase clarity, we have now listed the different investigated plant-based foodstuffs and environmental matrices (Results → Research trends):

Furthermore, during the study period, the majority of investigations into food products concentrated on animal-origin products whereas plant-based foods (including fruits, vegetables, spices/herbs, and grains) accounted for 5.6% of the examined foodstuffs. Finally, across the selected publications, seven environmental matrices (including food and processing plants, public lavatory, sandbox, slaughter knife, soil, and water) and 21 invertebrate taxa (mosquitoes: 47.8%; ticks; 39.1%; sand flies, gastropods, and fleas: 4.3% each) were investigated.

- In the conclusion, “multidisciplinary data” has been replaced by “multi-source data”.

- Figure 1, we have added specification on the figure:

“Solid lines show fitted trend (loess regression); shaded areas represent the corresponding 95% confidence interval.”

- The website of the Austrian Agency for Health and Food Safety (AGES) has been restructured and the updated link to the database has been added to the manuscript.
- Some typological errors have also been corrected.

REVIEWER COMMENTS

Reviewer #1 (Remarks to the Author):

Thanks to the authors who have answered to all my requests or concerns with many details, provided more explanations in the text and produced new analysis and figures to improve their work

I have no further comment

Reviewer #2 (Remarks to the Author):

Main concerns:

Although the authors addressed most of my previous comments well, I am still concerned about how network theory is used and how network properties are interpreted in this study.

In particular, although I understand what the authors intended to convey, I am not yet convinced about the use of the terms 'direct co-sharing' and 'indirect co-sharing', as well as their meanings within the network.

First, the term 'direct co-sharing' could be misleading because detecting a zoonotic agent in two different sources does not necessarily indicate transmission between those sources. The term could be interpreted as implying there is direct epidemiological evidence linking the sources.

Regarding 'indirect co-sharing', two sources might be considered indirectly connected even if no zoonotic agents are found to be shared between them. For example, consider sources A, B, C, and D, and agents i, j, and k. If agent i is found in sources A and B, agent j in B and C, and agent k in C and D, then in your network, A and D would be considered connected even though they share no agents. Furthermore, what if agents i and j are totally different, for instance, a bacteria and a virus, or even among bacteria, what if they are from two different families or phyla?

There would be other contexts that could complicate the interpretation of those links. What does this indirect connection between A and D represent? I appreciate that the authors highlighted this limitation by using "transmission-potential network" in the revised manuscript. Still, I am sceptical if that change in wording could resolve my (or potentially readers') concerns.

My concerns extend to the use of closeness and betweenness, which depend on indirect connections between actors in the network. The authors wrote, "share numerous zoonotic agents with numerous sources would have high closeness centrality." Although closeness is a well-known network property, what biological or epidemiological meaning does your result have? Also, the authors indicated that sources with high betweenness "may act as bridges between host communities (without necessarily transmitting zoonotic agents across these communities)." What do these bridges then indicate if they do not necessarily transmit zoonotic agents (for example, considering the reason I wrote above)? For me, network properties only with topological relationships are hard to interpret.

Specific comments:

Line 147: It may be helpful to explain why the non-detection of Usutu virus (USUV) is surprising, for those unfamiliar with the virus and its epidemiological context in Austria.

Line 201: Degree and strength centrality should reflect "the extent of direct co-sharing of zoonotic agents among sources," rather than merely "co-sharing patterns."

Line 225: Although this study focuses on zoonotic agents, communities other than community 1 do not include Homo sapiens. That is, despite the detection of a variety of zoonotic agents in those

communities, none has been detected in humans in Austria. It would be beneficial to discuss what this implies.

Line 288: "This outcome underscores an imbalance in research interest." Could this be considered a sign of imbalance? I would agree if some zoonotic agents were researched more extensively than warranted by their incidence and consequences. If there were few investigations of helminth infections and these infections are uncommon in Austria, could we then say that research into these infections is imbalanced?

Line 362: I assume this part is based on the six cliques ranked according to edge width. I would interpret this as evidence for the co-occurrence of zoonotic agents shared among actors in the clique, rather than "an increased probability of zoonotic spillover," since no formal risk assessment regarding this likelihood was conducted in this study.

Reviewer #3 (Remarks to the Author):

the authors have uptake all of my remarks and suggestions to improve their manuscript, which is I believe now fit for publication.

Response to Reviewers

Reviewer #1

Reviewer #1 (Remarks to the Author):

Thanks to the authors who have answered to all my requests or concerns with many details, provided more explanations in the text and produced new analysis and figures to improve their work.

I have no further comment.

Thank you, Reviewer #1, for your thorough review and appreciation of our efforts. We are grateful for your positive feedback.

Reviewer #2

Reviewer #2 (Remarks to the Author):

Main concerns:

Although the authors addressed most of my previous comments well, I am still concerned about how network theory is used and how network properties are interpreted in this study.

In particular, although I understand what the authors intended to convey, I am not yet convinced about the use of the terms 'direct co-sharing' and 'indirect co-sharing', as well as their meanings within the network.

First, the term 'direct co-sharing' could be misleading because detecting a zoonotic agent in two different sources does not necessarily indicate transmission between those sources. The term could be interpreted as implying there is direct epidemiological evidence linking the sources.

Regarding 'indirect co-sharing', two sources might be considered indirectly connected even if no zoonotic agents are found to be shared between them. For example, consider sources A, B, C, and D, and agents i, j, and k. If agent i is found in sources A and B, agent j in B and C, and agent k in C and D, then in your network, A and D would be considered connected even though they share no agents. Furthermore, what if agents i and j are totally different, for instance, a bacteria and a virus, or even among bacteria, what if they are from two different families or phyla?

There would be other contexts that could complicate the interpretation of those links. What does this indirect connection between A and D represent? I appreciate that the authors highlighted this limitation by using "transmission-potential network" in the revised manuscript. Still, I am sceptical if that change in wording could resolve my (or potentially readers') concerns.

My concerns extend to the use of closeness and betweenness, which depend on indirect connections between actors in the network. The authors wrote, "share numerous zoonotic agents with numerous sources would have high closeness centrality." Although closeness is a well-known network property, what biological or epidemiological meaning does your result have? Also, the authors indicated that sources with high betweenness "may act as bridges between host communities (without necessarily transmitting zoonotic agents across these communities)." What do these bridges then indicate if they do not necessarily transmit

zoonotic agents (for example, considering the reason I wrote above)? For me, network properties only with topological relationships are hard to interpret.

We thank Reviewer #2 for sharing his concerns about the method and interpretation of the analyses.

This study leverages established network methods and concepts successfully applied in ecology and epidemiology. Graph theory provides a robust and well-formalised framework to handle and interpret interactions between species ^{1,2}. To further contextualize our work, we reference Bascompte ² who explores the widespread application of ecological networks and emphasizes the inherent diversity of species and their interactions.

Qualitative bipartite networks are useful in describing the complex interactions between sets of species. However, they are not always the most convenient to analyse. To address this issue, one-mode projections of bipartite networks³ are commonly employed^{1,4-6}, where a link is created between two entities when they share something (e.g., a prey, a parasite, see Figure below). This approach is typically used in host-pathogen (or parasite) network analysis^{3,7-14}. Yet, these projections inevitably discard some information that is present in the original bipartite network, and therefore, display a “simplified” version of it. For example, in our study, the projection “loses” information on which zoonotic agents two sources have in common. Nevertheless, this information can be easily retrieved through our new interactive visualization.

The concern raised regarding interpretation of the topology of the network is inherent to any one-mode projection of a bipartite network. Other examples include the scientific collaboration network¹⁵ or the network of actors and films. The one-mode projection of the actor-film network creates an edge between two films (or actors) when they share one or more actors (or films); consequently, films (actors) that do not share any actors (films) may be indirectly connected ^{3,16}.

Despite this limitation, one-mode projections offer significant advantages. Their simplified structure facilitates analysis, particularly for tasks like community detection ^{3,6}. In our study, we partially mitigate the information loss by assigning weights to edges between sources, representing the number of shared zoonotic agents corrected by the research effort.

Figure: Using network approaches to understand interactions in host–parasite networks, from Runghen, et al. ⁵.

Editorial Note: Figure above reproduced from Runghen, R., Poulin, R., Monlleó-Borrull, C. & Llopis-Belenguer, C. Network Analysis: ten years shining light on host–parasite interactions. *Trends Parasitol.* **37**, 445-455 (2021), with permissions from Elsevier. © 2021 Elsevier Ltd. All rights reserved.

We agree that the term “direct co-sharing”, although previously employed by some authors⁹, could be misleading. We thank Reviewer #2 for this insightful comment. We have revised this term to “co-occurrence pattern”. This change ensures that the sentence does not imply direct transmission.

We also agree that “indirect co-sharing”, although previously employed by some authors⁹, could be confusing. To reflect the existence of indirect (ecological or epidemiological) links among sources, we have preferred the terms “indirect interactions”, as mentioned in Simmons, et al. ⁶, and have added this reference.

Revised sentence:

Degree and strength centrality reflect co-occurrence pattern of zoonotic agents among sources⁸. In contrast, betweenness and closeness centrality provide insights into indirect interactions through other sources⁶.

Analysis of host-parasite (or pathogen) networks can involve multiple parasite/pathogen taxa and multiple host taxa, as demonstrated by previous studies ^{9,12,13,17-19}. This approach proves particularly valuable in systems characterised by diverse hosts and infectious agents, where representing the complete set of interactions allows to characterise key aspects of zoonotic transmission, identify zoonotic reservoirs, and predict future zoonotic interactions. We view this multi-taxa approach as a strength, not a limitation, of our study. For instance, in the case of *Toxoplasma*, the transmission cycle may involve several sources, such as ingesting an intermediate animal host or ingesting environmental oocysts, with different host taxa involved. Therefore, our network design offers a relevant epidemiological model for examining complex multi-source multi-agent systems^{9,19}. Following Reviewer #2’s comment, and to better reflect potential barriers in the transmission chain, we have revised a sentence in our discussion, adding the notion of immunological barriers:

*However, these indirect interactions may lack epidemiological significance if, for instance, **immunological or physical barriers prevent agent transfer between sources, such as when the sources do not share similar ecological niche.***

The concept of “transmission-potential network” has been acknowledged in the literature, see Pilosof, et al. ⁹. It has been used in multi-host networks based on parasite sharing, and we believe it accurately reflects the concern raised by Reviewer #2 on indirect links. We have placed the reference immediately after the term to improve reader comprehension.

We used four centrality metrics as these measures are complementary^{6,8}, measuring different aspects of the node importance in the network, and must be interpreted within the specific system studied. Here, we did not aim to revisit the definition of closeness or betweenness, defined for weighted networks^{3,20}. These measures have been largely applied to ecological networks, including networks of parasite sharing, and the literature provides consistent, robust definitions^{1,3,7,10,13,14,21}, that we have used in our paper. For example, in Delmas, et al. ¹ (this reference has been included in the manuscript to enhance clarity and provide a clear point of reference for the reader.):

“Closeness centrality (CC) measures the proximity of a species to all other species in the network, and is therefore global in that, although defined at the species level, it accounts for the structure of the entire network. It is based on the shortest path length between pairs of species and thus indicates how rapidly/efficiently a node is likely to influence the overall network. The node with the highest CC is closer to

all other nodes than any other nodes and will thus affect more rapidly the overall network if, for example, there is a perturbation”.

“Betweenness centrality (CB) describes the number of times a species is between a pair of other species, i.e., how many paths (either directed or not) go through it. This measure is thus ideal to study the influence of species loss on fragmentation processes (...). Nodes with high CB values are considered as module connectors in modular networks.”

Following Reviewer #2 comment, we have removed the sentence in brackets *“(without necessarily transmitting zoonotic agents across these communities)”* as it was confusing and did not accurately convey the potential epidemiological role of the nodes with high betweenness. We also now refer to the paper of Granovetter ²² to fully explain the notion of bridge and the importance of indirect connections in the network. In this paper, Granovetter ²² introduced the theory of “weak ties”, which explains how social connections can influence information flow. In social networks, strong ties (i.e., close relationships) provide crucial emotional support but limited new information whereas weak ties (i.e., casual acquaintances) connect individuals to different social circles, offering access to novel and diverse information, which can be helpful in areas like finding a job or exploring new opportunities. Weak ties play a crucial structural role in social networks, they function as “bridges”, connecting otherwise isolated clusters of people within the network. Therefore, in a network, nodes that lies frequently on many different indirect connections between other nodes (i.e., having high “betweenness”) are more likely to act as bridges²³.

Finally, the practical use of nodes (sources) with high betweenness as “cut-points” in the network was discussed in the context of outbreak control in our Discussion. To clarify this point, we have revised the sentence as follows:

Nodes to be prioritised for surveillance may be different than those used for disease control. Removing central nodes in the network, e.g., via vaccination or culling targeting “bridge” zoonotic sources (i.e., with high betweenness), can significantly reduce the connectivity of the zoonotic web.

Specific comments:

- Line 147: It may be helpful to explain why the non-detection of Usutu virus (USUV) is surprising, for those unfamiliar with the virus and its epidemiological context in Austria.

Thank you for noticing. We have revised the sentence as follows:

Surprisingly, despite detection of Usutu virus (USUV) in various bird species, horses, and humans across the reviewed studies, it was not reported in arthropod vectors, a necessary component of its biological cycle (Supplementary Table 3).

- Line 201: Degree and strength centrality should reflect “the extent of direct co-sharing of zoonotic agents among sources,” rather than merely “co-sharing patterns.”

We agree with the comment. After careful consideration and based on the general comments and suggestions (above), we have opted for the term “co-occurrence pattern” in this specific sentence and excluded the term of “co-sharing”. The revised sentence is as follows:

Degree and strength centrality reflect co-occurrence pattern of zoonotic agents among sources.

We believe this phrasing maintains clarity while aligning with an accurate terminology as much as possible.

- Line 225: Although this study focuses on zoonotic agents, communities other than community 1 do not include Homo sapiens. That is, despite the detection of a variety of zoonotic agents in those communities, none has been detected in humans in Austria. It would be beneficial to discuss what this implies.

The Leiden algorithm used in this study, only considers discrete, non-overlapping communities, and therefore attribute each node to one community only²⁴. Other algorithms exist to detect overlapping communities^{25,26} but were not used here.

Our network displays the zoonotic ecosystem, where all infectious agents in the bipartite network have the potential to infect humans, and all agents shared within the unipartite network of agent sharing are zoonotic. The objective of the community detection was to pinpoint which species share the highest number of infectious agents with human, thus highlighting potential areas of high risk for zoonotic spillover, while also exploring other communities where zoonotic agents were circulating. To enhance clarity, we have added a sentence into the Discussion section:

*Our results indicate that the community including human, the oldest domesticated species (e.g., dog, cat, sheep, cattle, pig), and synanthropic species (e.g., Norway rat, house mouse) shares the most zoonotic agents. **This suggests that the highest risk of zoonotic spillover originates from sources within this community.***

- Line 288: “This outcome underscores an imbalance in research interest.” Could this be considered a sign of imbalance? I would agree if some zoonotic agents were researched more extensively than warranted by their incidence and consequences. If there were few investigations of helminth infections and these infections are uncommon in Austria, could we then say that research into these infections is imbalanced?

Thank you for this comment.

We respectfully disagree and believe that some zoonotic agents receive extensive research attention despite their relatively low incidence in the country. For instance, in Austria, the annual incidence of West Nile virus (WNV) in humans is relatively low, with 6 to 21 cases reported annually between 2018 and 2022²⁷. However, there is a legal requirement for the surveillance of WNV in the country and surveillance is implemented in mosquitoes, wild birds, and horses (<https://www.ages.at/en/human/disease/pathogens-from-a-to-z/west-nile-virus>). Consequently, WNV has been the subject of investigations in 57 animal hosts across 16 studies, ranking 5th in publication frequency. In contrast, Austria does not have surveillance system for toxoplasmosis²⁷. Yet, there were an estimated 647 congenital toxoplasmosis cases in 2010^{28,29} (no updated data available). Despite the public health importance of this parasite, the possible long-term consequences of congenital toxoplasmosis, and the knowledge gap on the transmission sources, especially to pregnant women, only 24 studies have been conducted on toxoplasmosis in Austria, none of them have investigated environmental or food matrices.

These disparities highlight disproportionate research focus between notifiable diseases like WNV and non-notifiable ones like toxoplasmosis. Furthermore, helminth infections are typically under-reported in Europe³⁰⁻³³. Although we provide here several references for our statement, only one was added to the manuscript to limit the number of citations, which is already higher than recommended in the Authors’ Guidelines.

Moreover, we would not use the word “uncommon” to describe the importance of a disease as the potential impact of a disease is not solely determined by its incidence. It is crucial to recognize that the costs associated with treating and managing each case, as well as the loss of life-days, may be considerable.

- Line 362: I assume this part is based on the six cliques ranked according to edge width. I would interpret this as evidence for the co-occurrence of zoonotic agents shared among actors in the clique, rather than “an increased probability of zoonotic spillover,” since no formal risk assessment regarding this likelihood was conducted in this study.

Thank you for this comment. We believe it refers to line 333. We agree with the proposed terminology as it better reflects the analysis performed. Therefore, we have revised the sentence as follows:

Our findings demonstrate that there is an increased co-occurrence of zoonotic agents at human-cattle and human-food interfaces, suggesting an elevated likelihood of zoonotic spillover.

Reviewer #3

Reviewer #3 (Remarks to the Author):

the authors have uptake all of my remarks and suggestions to improve their manuscript, which is I believe now fit for publication.

Thank you, Reviewer #3, for your positive feedback. We're glad to hear that you find our paper suitable for publication.

Other changes

In Table 1, “investigations” was replaced by “detections” as this table only refers to (sero)positive results (investigations is used in the Methods section to indicate positive and negative results).

Additionally, we have improved the language in two sentences by making slight modifications without altering their meaning.

Some typological errors have been corrected.

References used in this Response

- 1 Delmas, E. *et al.* Analysing ecological networks of species interactions. *Biological Reviews* **94**, 16-36 (2019).
- 2 Bascompte, J. Disentangling the web of life. *Science* **325**, 416-419 (2009).
- 3 Newman, M. *Networks: An Introduction*. 2nd edn, (Oxford University Press, Inc., 2018).
- 4 Montoya, J. M., Pimm, S. L. & Solé, R. V. Ecological networks and their fragility. *Nature* **442**, 259-264 (2006).
- 5 Runghen, R., Poulin, R., Monlleó-Borrull, C. & Llopis-Belenguer, C. Network Analysis: ten years shining light on host–parasite interactions. *Trends Parasitol.* **37**, 445-455 (2021).

- 6 Simmons, B. I. *et al.* Motifs in bipartite ecological networks: uncovering indirect interactions. *Oikos* **128**, 154-170 (2019).
- 7 Luis, A. D. *et al.* Network analysis of host–virus communities in bats and rodents reveals determinants of cross-species transmission. *Ecol. Lett.* **18**, 1153-1162 (2015).
- 8 Gómez, J. M., Nunn, C. L. & Verdú, M. Centrality in primate–parasite networks reveals the potential for the transmission of emerging infectious diseases to humans. *PNAS* **110**, 7738-7741 (2013).
- 9 Pilosof, S., Morand, S., Krasnov, B. R. & Nunn, C. L. Potential parasite transmission in multi-host networks based on parasite sharing. *PLoS One* **10**, e0117909 (2015).
- 10 Pilosof, S., Fortuna, M. A., Vinarski, M. V., Korallo-Vinarskaya, N. P. & Krasnov, B. R. Temporal dynamics of direct reciprocal and indirect effects in a host–parasite network. *Journal of Animal Ecology* **82**, 987-996 (2013).
- 11 Poulin, R. Network analysis shining light on parasite ecology and diversity. *Trends Parasitol.* **26**, 492-498 (2010).
- 12 Wardeh, M., Sharkey, K. J. & Baylis, M. Integration of shared-pathogen networks and machine learning reveals the key aspects of zoonoses and predicts mammalian reservoirs. *Proceedings of the Royal Society B: Biological Sciences* **287**, 20192882 (2020).
- 13 Dallas, T. A. *et al.* Host traits associated with species roles in parasite sharing networks. *Oikos* **128**, 23-32 (2019).
- 14 VanderWaal, K. L., Atwill, E. R., Isbell, L. A. & McCowan, B. Quantifying microbe transmission networks for wild and domestic ungulates in Kenya. *Biol. Conserv.* **169**, 136-146 (2014).
- 15 Newman, M. E. J. Scientific collaboration networks. II. Shortest paths, weighted networks, and centrality. *Phys. Rev. E* **64**, 016132 (2001).
- 16 Hopkins, B. Kevin bacon and graph theory. *Problems, Resources, and Issues in Mathematics Undergraduate Studies* **14**, 5-11 (2004).
- 17 Griffiths, E. C., Pedersen, A. B., Fenton, A. & Petchey, O. L. Analysis of a summary network of co-infection in humans reveals that parasites interact most via shared resources. *Proceedings of the Royal Society B: Biological Sciences* **281**, 20132286 (2014).
- 18 Mouillot, D., Krasnov, B. R. & Poulin, R. High intervality explained by phylogenetic constraints in host–parasite webs. *Ecology* **89**, 2043-2051 (2008).
- 19 Lima Jr, D. P., Giacomini, H. C., Takemoto, R. M., Agostinho, A. A. & Bini, L. M. Patterns of interactions of a large fish–parasite network in a tropical floodplain. *Journal of Animal Ecology* **81**, 905-913 (2012).
- 20 Opsahl, T., Agneessens, F. & Skvoretz, J. Node centrality in weighted networks: Generalizing degree and shortest paths. *Social Networks* **32**, 245-251 (2010).
- 21 Martín González, A. M., Dalsgaard, B. & Olesen, J. M. Centrality measures and the importance of generalist species in pollination networks. *Ecological Complexity* **7**, 36-43 (2010).
- 22 Granovetter, M. S. The strength of weak ties. *Am. J. Sociol.* **78**, 1360-1380 (1973).
- 23 Marsden, P. V. & Campbell, K. E. Reflections on conceptualizing and measuring tie strength. *Social Forces* **91**, 17-23 (2012).
- 24 Traag, V. A., Waltman, L. & van Eck, N. J. From Louvain to Leiden: guaranteeing well-connected communities. *Sci. Rep.* **9**, 5233 (2019).
- 25 Fortunato, S. & Newman, M. E. J. 20 years of network community detection. *Nature Physics* **18**, 848-850 (2022).
- 26 Palla, G., Derényi, I., Farkas, I. & Vicsek, T. Uncovering the overlapping community structure of complex networks in nature and society. *Nature* **435**, 814-818 (2005).
- 27 European Food Safety Authority & European Centre for Disease Prevention. The European Union One Health 2022 Zoonoses Report. *EFSA Journal* **21**, p211202 (2023).
- 28 Edelhofer, R. & Prossinger, H. Infection with *Toxoplasma gondii* during Pregnancy: Seroepidemiological Studies in Austria. *Zoonoses and Public Health* **57**, 18-26 (2010).

- 29 Statistik Austria. *Demographic characteristics of newborns*, <https://www.statistik.at/en/statistics/population-and-society/population/births/demographic-characteristics-of-newborns> (2024).
- 30 Casulli, A. *et al.* Unveiling the incidences and trends of the neglected zoonosis cystic echinococcosis in Europe: a systematic review from the MEmE project. *Lancet Infect. Dis.* **23**, e95-e107 (2023).
- 31 Jorgensen, P. *et al.* Underreporting of human alveolar echinococcosis, Germany. *Emerg. Infect. Dis.* **14**, 935-937 (2008).
- 32 Trevisan, C. *et al.* Epidemiology of taeniosis/cysticercosis in Europe, a systematic review: eastern Europe. *Parasites & Vectors* **11**, 569 (2018).
- 33 Lassen, B. *et al.* Serological evidence of exposure to globally relevant zoonotic parasites in the Estonian population. *PLoS One* **11**, e0164142 (2016).

REVIEWERS' COMMENTS

Reviewer #2 (Remarks to the Author):

I thank the authors for answering my comments in significant details and for addressing these in the main text.

I have no further comment.